# Shape As Points: A Differentiable Poisson Solver

**Songyou Peng**[1,2]    **Chiyu "Max" Jiang**[*][†]    **Yiyi Liao**[2,3][†]    **Michael Niemeyer**[2,3]

**Marc Pollefeys**[1,4]    **Andreas Geiger**[2,3]

[1]ETH Zurich        [2]Max Planck Institute for Intelligent Systems, Tübingen
[3]University of Tübingen        [4]Microsoft

## Abstract

In recent years, neural implicit representations gained popularity in 3D reconstruction due to their expressiveness and flexibility. However, the implicit nature of neural implicit representations results in slow inference time and requires careful initialization. In this paper, we revisit the classic yet ubiquitous point cloud representation and introduce a differentiable point-to-mesh layer using a differentiable formulation of Poisson Surface Reconstruction (PSR) that allows for a GPU-accelerated fast solution of the indicator function given an oriented point cloud. The differentiable PSR layer allows us to efficiently and differentiably bridge the explicit 3D point representation with the 3D mesh via the implicit indicator field, enabling end-to-end optimization of surface reconstruction metrics such as Chamfer distance. This duality between points and meshes hence allows us to represent shapes as oriented point clouds, which are explicit, lightweight and expressive. Compared to neural implicit representations, our Shape-As-Points (SAP) model is more interpretable, lightweight, and accelerates inference time by one order of magnitude. Compared to other explicit representations such as points, patches, and meshes, SAP produces topology-agnostic, watertight manifold surfaces. We demonstrate the effectiveness of SAP on the task of surface reconstruction from unoriented point clouds and learning-based reconstruction.

## 1 Introduction

Shape representations are central to many of the recent advancements in 3D computer vision and computer graphics, ranging from neural rendering [41,45,48,55,58] to shape reconstruction [10,26,40, 47,50,52,70]. While conventional representations such as point clouds and meshes are efficient and well-studied, they also suffer from several limitations: Point clouds are lightweight and easy to obtain, but do not directly encode surface information. Meshes, on the other hand, are usually restricted to fixed topologies. More recently, neural implicit representations [10,40,50] have shown promising results for representing geometry due to their flexibility in encoding varied topologies, and their easy integration with differentiable frameworks. However, as such representations implicitly encode surface information, extracting the underlying surface is typically slow as they require numerous network evaluations in 3D space for extracting complete surfaces using marching cubes [10,40,50], or along rays for intersection detection in the context of volumetric rendering [45,47,49,70].

In this work, we introduce a novel Poisson solver which performs fast GPU-accelerated Differentiable Poisson Surface Reconstruction (DPSR) and solves for an indicator function from an oriented point cloud in a few milliseconds. Thanks to the differentiablility of our Poisson solver, gradients from a loss on the output mesh or a loss on the intermediate indicator grid can be efficiently backpropagated to update the oriented point cloud representation. This differential bridge between points, indicator

---

[*]Work done while at UC Berkeley.
[†]Corresponding authors.

35th Conference on Neural Information Processing Systems (NeurIPS 2021).

| Representations | | Points [17] | Voxels [11] | Meshes [61] | Patches [20] | Implicits [40] | **SAP** (Ours) | GT |
|---|---|---|---|---|---|---|---|---|
| Efficiency | Grid Eval Time ($128^3$) | n/a | n/a | n/a | n/a | 0.33s | 0.012s | |
| Priors | Easy Initialization | ✔ | ✔ | ✔ | ✗ | ✗ | ✔ | |
| | Watertight | ✗ | ✔ | ✔ | ✗ | ✔ | ✔ | |
| Quality | No Self-intersection | n/a | n/a | ✗ | ✗ | ✔ | ✔ | |
| | Topology-Agnostic | ✔ | ✔ | ✗ | ✔ | ✔ | ✔ | |

Table 1: **Overview of Different Shape Representations.** Shape-As-Points produces higher quality geometry compared to other explicit representations [11, 17, 20, 61] and requires significantly less inference time for extracting geometry compared to neural implicit representations [40].

functions, and meshes allows us to represent shapes as oriented point clouds. We therefore call this shape representation *Shape-As-Points* (SAP). Compared to existing shape representations, Shape-As-Points has the following advantages (see also Table 1):

**Efficiency:** SAP has a low memory footprint as it only requires storing a collection of oriented point samples at the surface, rather than volumetric quantities (voxels) or a large number of network parameters for neural implicit representations. Using spectral methods, the indicator field can be computed efficiently (12 ms at $128^3$ resolution[3]), compared to the typical rather slow query time of neural implicit networks (330 ms using [40] at the same resolution). **Accuracy:** The resulting mesh can be generated at high resolutions, is guaranteed to be watertight, free from self-intersections and also topology-agnostic. **Initialization:** It is easy to initialize SAP with a given geometry such as template shapes or noisy observations. In contrast, neural implicit representations are harder to initialize, except for few simple primitives like spheres [1]. See supplementary for more discussions.

To investigate the aforementioned properties, we perform a set of controlled experiments. Moreover, we demonstrate state-of-the-art performance in reconstructing surface geometry from unoriented point clouds in two settings: an optimization-based setting that does not require training and is applicable to a wide range of shapes, and a learning-based setting for conditional shape reconstruction that is robust to noisy point clouds and outliers. In summary, the main contributions of this work are:

- We present Shape-As-Points, a novel shape representation that is interpretable, lightweight, and yields high-quality watertight meshes at low inference times.

- The core of the Shape-As-Points representation is a versatile, differentiable and generalizable Poisson solver that can be used for a range of applications.

- We study various properties inherent to the Shape-As-Points representation, including inference time, sensitivity to initialization and topology-agnostic representation capacity.

- We demonstrate state-of-the-art reconstruction results from noisy unoriented point clouds at a significantly reduced computational budget compared to existing methods.

Code is available at https://github.com/autonomousvision/shape_as_points.

## 2 Related Work

### 2.1 3D Shape Representations

3D shape representations are central to 3D computer vision and graphics. Shape representations can be generally categorized as being either *explicit* or *implicit*. *Explicit* shape representations and learning algorithms depending on such representations directly parameterize the surface of the geometry, either as a point cloud [17, 38, 53, 54, 64, 67], parameterized mesh [22, 24, 27, 61] or surface patches [2, 20, 37, 43, 65, 68, 69]. Explicit representations are usually lightweight and require few parameters to represent the geometry, but suffer from discretization, the difficulty to represent

---

[3]On average, our method requires 12 ms for computing a $128^3$ indicator grid from 15K points on a single NVIDIA GTX 1080Ti GPU. Computing a $256^3$ indicator grid requires 140 ms.

watertight surfaces (point clouds, surface patches), or are restricted to a pre-defined topology (mesh). *Implicit* representations, in contrast, represent the shape as a level set of a continuous function over a discretized voxel grid [14, 25, 33, 66] or more recently parameterized as a neural network, typically referred to as neural implicit functions [10, 40, 50]. Neural implicit representations have been successfully used to represent geometries of objects [10, 16, 18, 35, 40, 44, 47, 50, 57, 59, 62, 63] and scenes [8, 26, 34, 46, 52, 57]. Additionally, neural implicit functions are able to represent radiance fields which allow for high-fidelity appearance and novel view synthesis [39, 45]. However, extracting surface geometry from implicit representations typically requires dense evaluation of multi-layer perceptrons, either on a volumetric grid or along rays, resulting in slow inference time. In contrast, SAP efficiently solves the Poisson Equation during inference by representing the shape as an oriented point cloud.

### 2.2 Optimization-based 3D Reconstruction from Point Clouds

Several works have addressed the problem of inferring continuous surfaces from a point cloud. They tackle this task by utilizing basis functions, set properties of the points, or neural networks. Early works in shape reconstruction from point clouds utilize the convex hull or alpha shapes for reconstruction [15]. The ball pivoting algorithm [5] leverages the continuity property of spherical balls of a given radius. One of the most popular techniques, Poisson Surface Reconstruction (PSR) [28, 29], solves the Poisson Equation and inherits smoothness properties from the basis functions used in the Poisson Equation. However, PSR is sensitive to the normals of the input points which must be inferred using a separate preprocessing step. In contrast, our method does not require any normal estimation and is thus more robust to noise. More recent works take advantage of the continuous nature of neural networks as function approximators to fit surfaces to point sets [19, 23, 42, 65]. However, these methods tend to be memory and computationally intensive, while our method yields high-quality watertight meshes in a few milliseconds.

### 2.3 Learning-based 3D Reconstruction from Point Clouds

Learning-based approaches exploit a training set of 3D shapes to infer the parameters of a reconstruction model. Some approaches focus on local data priors [2, 26] which typically result in better generalization, but suffer when large surfaces must be completed. Other approaches learn object-level [33, 40, 50] or scene-level priors [12, 13, 26, 52]. Most reconstruction approaches directly reconstruct a meshed surface geometry, though some works [3, 4, 21, 31] first predict point set normals to subsequently reconstruct the geometry via PSR [28, 29]. However, such methods fail to handle large levels of noise, since they are unable to move points or selectively ignore outliers. In contrast, our end-to-end approach is able to address this issue by either moving outlier points to the actual surface or by selectively muting outliers either by forming paired point clusters that self-cancel or reducing the magnitude of the predicted normals which controls their influence on the reconstruction.

## 3 Method

At the core of the Shape-As-Points representation is a differentiable Poisson solver, which can be used for both optimization-based and learning-based surface estimation. We first introduce the Poisson solver in Section 3.1. Next, we investigate two applications using our solver: optimization-based 3D reconstruction (Section 3.2) and learning-based 3D reconstruction (Section 3.3).

### 3.1 Differentiable Poisson Solver

The key step in Poisson Surface Reconstruction [28, 29] involves solving the Poisson Equation. Let $\mathbf{x} \in \mathbb{R}^3$ denote a spatial coordinate and $\mathbf{n} \in \mathbb{R}^3$ denote its corresponding normal. The Poisson Equation arises from the insight that a set consisting of point coordinates and normals $\{\mathbf{p} = (\mathbf{c}, \mathbf{n})\}$ can be viewed as samples of the gradient of the underlying implicit indicator function $\chi(\mathbf{x})$ that describes the solid geometry. We define the normal vector field as a superposition of pulse functions $\mathbf{v}(\mathbf{x}) = \sum_{(\mathbf{c}_i, \mathbf{n}_i) \in \{\mathbf{p}\}} \delta(\mathbf{x} - \mathbf{c}_i, \mathbf{n}_i)$, where $\delta(\mathbf{x}, \mathbf{n}) = \{\mathbf{n}$ if $\mathbf{x} = 0$ and $0$ otherwise$\}$. By applying the divergence operator, the variational problem transforms into the standard Poisson equation:

$$\nabla^2 \chi := \nabla \cdot \nabla \chi = \nabla \cdot \mathbf{v} \tag{1}$$

In order to solve this set of linear Partial Differential Equations (PDEs), we discretize the function values and differential operators. Without loss of generality, we assume that the normal vector field $\mathbf{v}$ and the indicator function $\chi$ are sampled at $r$ uniformly spaced locations along each dimension. Denote the spatial dimensionality of the problem to be $d$. Without loss of generality, we consider

the three dimensional case where $n := r \times r \times r$ for $d = 3$. We have the indicator function $\chi \in \mathbb{R}^n$, the point normal field $\mathbf{v} \in \mathbb{R}^{n \times d}$, the gradient operator $\nabla : \mathbb{R}^n \mapsto \mathbb{R}^{n \times d}$, the divergence operator $(\nabla \cdot) : \mathbb{R}^{n \times d} \mapsto \mathbb{R}^n$, and the derived laplacian operator $\nabla^2 := \nabla \cdot \nabla : \mathbb{R}^n \mapsto \mathbb{R}^n$. Under such a discretization scheme, solving for the indicator function amounts to solving the linear system by inverting the divergence operator subject to boundary conditions of surface points having zero level set. Following [28], we fix the overall scale to $m = 0.5$ at $\mathbf{x} = 0$:

$$\chi = (\nabla^2)^{-1} \nabla \cdot \mathbf{v} \quad \text{s.t.} \quad \chi|_{\mathbf{x} \in \{\mathbf{c}\}} = 0 \quad \text{and} \quad \text{abs}(\chi|_{\mathbf{x}=0}) = m \tag{2}$$

**Point Rasterization:** We obtain the uniformly discretized point normal field $\mathbf{v}$ by rasterizing the point normals onto a uniformly sampled voxel grid. We can differentiably perform point rasterization via inverse trilinear interpolation, similar to the approach in [28, 29]. We scatter the point normal values to the voxel grid vertices, weighted by the trilinear interpolation weights. The point rasterization process has $\mathcal{O}(n)$ space complexity, linear with respect to the number of grid cells, and $\mathcal{O}(N)$ time complexity, linear with respect to the number of points. See supplementary for details.

**Spectral Methods for Solving PSR:** In contrast to the finite-element approach taken in [28, 29], we solve the PDEs using spectral methods [7]. While spectral methods are commonly used in scientific computing for solving PDEs and in some cases applied to computer vision problems [32], we are the first to apply them in the context of Poisson Surface Reconstruction. Unlike finite-element approaches that depend on irregular data structures such as octrees or tetrahedral meshes for discritizing space, spectral methods can be efficently solved over a uniform grid as they leverage highly optimized Fast Fourier Transform (FFT) operations that are well supported for GPUs, TPUs, and mainstream deep learning frameworks. Spectral methods decompose the original signal into a linear sum of functions represented using the sine / cosine basis functions whose derivatives can be computed analytically. This allows us to easily approximate differential operators in spectral space. We denote the spectral domain signals with a tilde symbol, i.e., $\tilde{\mathbf{v}} = \text{FFT}(\mathbf{v})$. We first solve for the unnormalized indicator function $\chi'$, not accounting for boundary conditions

$$\chi' = \text{IFFT}(\tilde{\chi}) \qquad \tilde{\chi} = \tilde{g}_{\sigma,r}(\mathbf{u}) \odot \frac{i\mathbf{u} \cdot \tilde{\mathbf{v}}}{-2\pi \|\mathbf{u}\|^2} \qquad \tilde{g}_{\sigma,r}(\mathbf{u}) = \exp\left(-2\frac{\sigma^2 \|\mathbf{u}\|^2}{r^2}\right) \tag{3}$$

where the spectral frequencies are denoted as $\mathbf{u} := (u, v, w) \in \mathbb{R}^{n \times d}$ corresponding to the $x, y, z$ spatial dimensions, and $\text{IFFT}(\tilde{\chi})$ represents the inverse fast Fourier transform of $\tilde{\chi}$. $\tilde{g}_{\sigma,r}(\mathbf{u})$ is a Gaussian smoothing kernel of bandwidth $\sigma$ at grid resolution $r$ in the spectral domain. The Gaussian kernel is used to mitigate ringing effects as a result of the Gibbs phenomenon from rasterizing the point normals. We denote the element-wise product as $\odot : \mathbb{R}^n \times \mathbb{R}^n \mapsto \mathbb{R}^n$, the L2-norm as $\|\cdot\|^2 : \mathbb{R}^{n \times d} \mapsto \mathbb{R}^n$, and the dot product as $(\cdot) : \mathbb{R}^{n \times d} \times \mathbb{R}^{n \times d} \mapsto \mathbb{R}^n$. Finally, we subtract by the mean of the indicator function at the point set and scale the indicator function to obtain the solution to the PSR problem in Eqn. 2:

$$\chi = \underbrace{\frac{m}{\text{abs}(\chi'|_{\mathbf{x}=0})}}_{\text{scale}} \underbrace{\left(\chi' - \frac{1}{|\{\mathbf{c}\}|} \sum_{\mathbf{c} \in \{\mathbf{c}\}} \chi'|_{\mathbf{x}=\mathbf{c}}\right)}_{\text{subtract by mean}} \tag{4}$$

A detailed derivation of our differentiable PSR solver is provided in the supplementary material.

### 3.2 SAP for Optimization-based 3D Reconstruction

We can use the proposed differentiable Poisson solver for various applications. First, we consider the classical task of surface reconstruction from unoriented point clouds. The overall pipeline for this setting is illustrated in Fig. 1 (top). We now provide details about each component.

**Forward pass:** It is natural to initialize the oriented 3D point cloud serving as 3D shape representation using the noisy 3D input points and corresponding (estimated) normals. However, to demonstrate the flexibility and robustness of our model, we purposefully initialize our model using a generic 3D sphere with radius $r$ in our experiments. Given the orientated point cloud, we apply our Poisson solver to obtain an indicator function grid, which can be converted to a mesh using Marching Cubes [36].

**Backward pass:** For every point $\mathbf{p}_{\text{mesh}}$ sampled from the mesh $\mathcal{M}$, we calculate a bi-directional L2 Chamfer Distance $\mathcal{L}_{\text{CD}}$ with respect to the input point cloud. To backpropagate the loss $\mathcal{L}_{\text{CD}}$ through

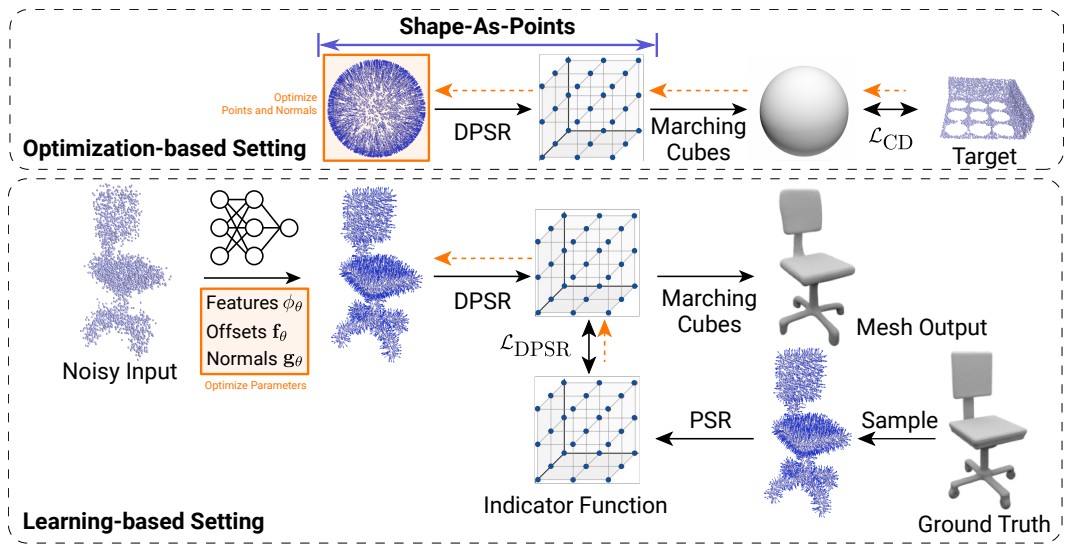

Figure 1: **Model Overview.** Top: Pipeline for optimization-based single object reconstruction. The Chamfer loss on the target point cloud is backpropagated to the source point cloud w/ normals for optimization. Bottom: Pipeline for learning-based surface reconstruction. Unlike the optimization-based setting, here we provide supervision at the indicator grid level, since we assume access to watertight meshes for supervision, as is common practice in learning-based single object reconstruction.

$\mathbf{p}_{\text{mesh}}$ to point $\mathbf{p}$ in our source oriented point cloud, we decompose the gradient using the chain rule:

$$\frac{\partial \mathcal{L}_{\text{CD}}}{\partial \mathbf{p}} = \frac{\partial \mathcal{L}_{\text{CD}}}{\partial \mathbf{p}_{\text{mesh}}} \frac{\partial \mathbf{p}_{\text{mesh}}}{\partial \chi} \frac{\partial \chi}{\partial \mathbf{p}} \tag{5}$$

All terms in (5) are differentialable except for the middle one $\frac{\partial \mathbf{p}_{\text{mesh}}}{\partial \chi}$ which involves Marching Cubes. However, this gradient can be effectively approximated by the inverse surface normal [56]:

$$\frac{\partial \mathbf{p}_{\text{mesh}}}{\partial \chi} = -\mathbf{n}_{\text{mesh}} \tag{6}$$

where $\mathbf{n}_{\text{mesh}}$ is the normal of the point $\mathbf{p}_{\text{mesh}}$. Different from MeshSDF [56] that uses the gradients to update the latent code of a pretrained implicit shape representation, our method updates the source point cloud using the proposed differentiable Poisson solver.

**Resampling:** To increase the robustness of the optimization process, we uniformly resample points and normals from the largest mesh component every 200 iterations, and replace all points in the original point clouds with the resampled ones. This resampling strategy eliminates outlier points that drift away during the optimization, and enforces a more uniform distribution of points. We provide an ablation study in supplementary.

**Coarse-to-fine:** To further decrease run-time, we consider a coarse-to-fine strategy during optimization. More specifically, we start optimizing at an indicator grid resolution of $32^3$ for 1000 iterations, from which we obtain a coarse shape. Next, we sample from this coarse mesh and continue optimization at a resolution of $64^3$ for 1000 iterations. We repeat this process until we reach the target resolution ($256^3$) at which we acquire the final output mesh. See also supplementary.

### 3.3 SAP for Learning-based 3D Reconstruction

We now consider the learning-based 3D reconstruction setting in which we train a conditional model that takes a noisy, unoriented point cloud as input and outputs a 3D shape. More specifically, we train the model to predict a clean oriented point cloud, from which we obtain a watertight mesh using our Poisson solver and Marching Cubes. We leverage the differentiability of our Poisson solver to learn the parameters of this conditional model. Following common practice, we assume watertight meshes as ground truth and consequently supervise directly with the ground truth indicator grid obtained

from these meshes. Fig. 1 (bottom) illustrates the pipeline of our architecture for the learning-based surface reconstruction task.

**Architecture:** We first encode the unoriented input point cloud coordinates $\{\mathbf{c}\}$ into a feature $\phi$. The resulting feature should encapsulate both local and global information about the input point cloud. We utilize the convolutional point encoder proposed in [52] for this purpose. Note that in the following, we will use $\phi_\theta(\mathbf{c})$ to denote the features at point $\mathbf{c}$, dropping the dependency of $\phi$ on the remaining points $\{\mathbf{c}\}$ for clarity. Also, we use $\theta$ to refer to network parameters in general.

Given their features, we aim to estimate both offsets and normals for every input point $\mathbf{c}$ in the point cloud $\{\mathbf{c}\}$. We use a shallow Multi-Layer Perceptron (MLP) $\mathbf{f}_\theta$ to predict the offset for $\mathbf{c}$:

$$\Delta\mathbf{c} = \mathbf{f}_\theta(\mathbf{c}, \phi_\theta(\mathbf{c})) \tag{7}$$

where $\phi(\mathbf{c})$ is obtained from the feature volume using trilinear interpolation. We predict $k$ offsets per input point, where $k \geq 1$. We add the offsets $\Delta\mathbf{c}$ to the input point position $\mathbf{c}$ and call the updated point position $\hat{\mathbf{c}}$. Additional offsets allow us to densify the point cloud, leading to enhanced reconstruction quality. We choose $k = 7$ for all learning-based reconstruction experiments (see ablation study in Table 4). For each updated point $\hat{\mathbf{c}}$, we use a second MLP $\mathbf{g}_\theta$ to predict its normal:

$$\hat{\mathbf{n}} = \mathbf{g}_\theta(\hat{\mathbf{c}}, \phi_\theta(\hat{\mathbf{c}})) \tag{8}$$

We use the same decoder architecture as in [52] for both $\mathbf{f}_\theta$ and $\mathbf{g}_\theta$. The network comprises 5 layers of ResNet blocks with a hidden dimension of 32. These two networks $\mathbf{f}_\theta$ and $\mathbf{g}_\theta$ do not share weights.

**Training and Inference:** During training, we obtain the estimated indicator grid $\hat{\chi}$ from the predicted point clouds $(\hat{\mathbf{c}}, \hat{\mathbf{n}})$ using our differentiable Poisson solver. Since we assume watertight and noise-free meshes for supervision, we acquire the ground truth indicator grid by running PSR on a densely sampled point clouds of the ground truth meshes with the corresponding ground truth normals. This avoids running Marching Cubes at every iteration and accelerates training. We use the Mean Square Error (MSE) loss on the predicted and ground truth indicator grid:

$$\mathcal{L}_{\text{DPSR}} = \|\hat{\chi} - \chi\|^2 \tag{9}$$

We implement all models in PyTorch [51] and use the Adam optimizer [30] with a learning rate of 5e-4. During inference, we use our trained model to predict normals and offsets, use DPSR to solve for the indicator grid, and run Marching Cubes [36] to extract meshes.

## 4 Experiments

Following the exposition in the previous section, we conduct two types of experiments to evaluate our method. First, we perform single object reconstruction from unoriented point clouds. Next, we apply our method to learning-based surface reconstruction on ShapeNet [9], using noisy point clouds with or without outliers as inputs.

**Datasets:** We use the following datasets for optimization-based reconstruction: 1) Thingi10K [71], 2) Surface reconstruction benchmark (SRB) [65] and 3) D-FAUST [6]. Similar to prior works, we use 5 objects per dataset [19, 23, 65]. For learning-based object-level reconstruction, we consider all 13 classes of the ShapeNet [9] subset, using the train/val/test split from [11].

**Baselines:** In the optimization-based reconstruction setting, we compare to network-based methods IGR [19] and Point2Mesh [23], as well as Screened Poisson Surface Reconstruction[4] (SPSR) [29] on plane-fitted normals. To ensure that the predicted normals are consistently oriented for SPSR, we propagate the normal orientation using the minimum spanning tree [72]. For learning-based surface reconstruction, we compare against point-based Point Set Generation Networks (PSGN) [17], patch-based AtlasNet [20], voxel-based 3D-R2N2 [11], and ConvONet [52], which has recently reported state-of-the-art results on this task. We use ConvOnet in their best-performing setting (3-plane encoders). SPSR is also used as a baseline. In addition, to evaluate the importance of our differentiable PSR optimization, we design another point-based baseline. This baseline uses the same network architecture to predict points and normals. However, instead of passing them to our Poisson solver and

---

[4] We use the official implementation https://github.com/mkazhdan/PoissonRecon.

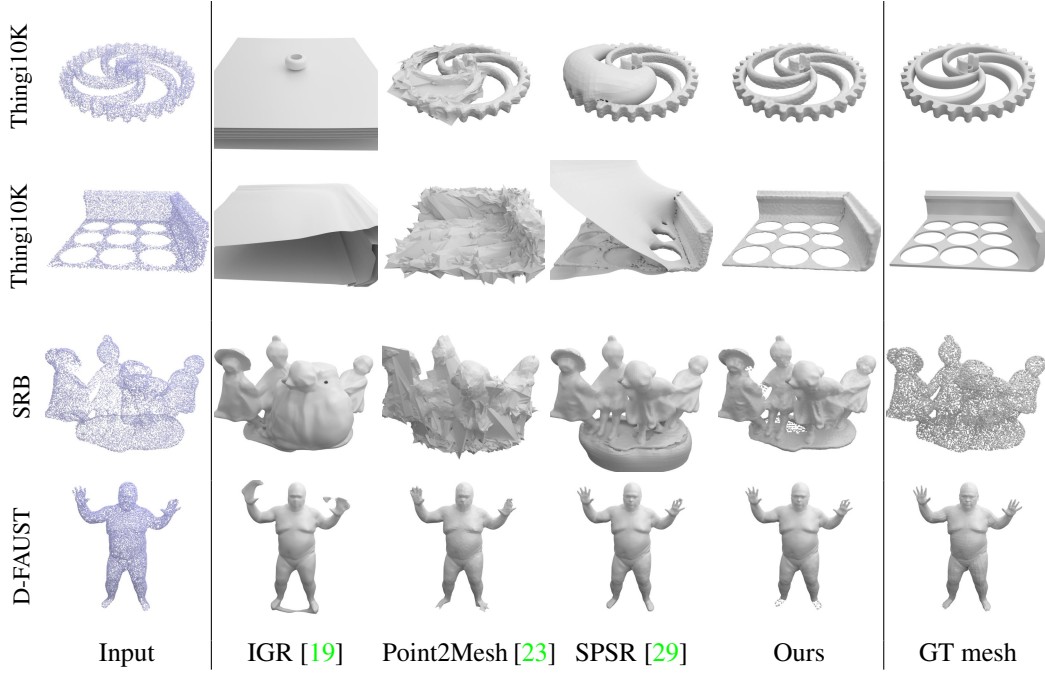

Figure 2: **Optimization-based 3D Reconstruction.** Input point clouds are downsampled for visualization. Note that the ground truth of SRB is provided as point clouds.

| Dataset | Method | Chamfer-$L_1$ ($\downarrow$) | F-Score ($\uparrow$) | Normal C. ($\uparrow$) | Time (s) |
|---------|--------|------------------|-----------|-----------------|----------|
| Thingi10K | IGR [19] | 0.440 | 0.505 | 0.692 | 1842.3 |
| | Point2Mesh [23] | 0.109 | 0.656 | 0.806 | 3714.7 |
| | SPSR [29] | 0.223 | 0.787 | 0.896 | 9.3 |
| | **Ours** | **0.054** | **0.940** | **0.947** | 370.1 |
| SRB | IGR [19] | 0.178 | 0.755 | – | 1847.6 |
| | Point2Mesh [23] | 0.116 | 0.648 | – | 4707.9 |
| | SPSR [29] | 0.232 | 0.735 | – | 9.2 |
| | **Ours** | **0.076** | **0.830** | – | 326.0 |
| D-FAUST | IGR [19] | 0.235 | 0.805 | 0.911 | 1857.2 |
| | Point2Mesh [23] | 0.071 | 0.855 | 0.905 | 3678.7 |
| | SPSR [29] | 0.044 | **0.966** | **0.965** | 4.3 |
| | **Ours** | **0.043** | **0.966** | 0.959 | 379.9 |

Table 2: **Optimization-based 3D Reconstruction**. Quantitative comparison on 3 datasets. Normal Consistency cannot be evaluated on SRB as this dataset provides only unoriented point clouds. Optimization time is evaluated on a single GTX 1080Ti GPU for IGR, Point2Mesh and our method.

calculate $\mathcal{L}_{\text{DPSR}}$ on the indicator grid, we directly supervise the point positions with a bi-directional Chamfer distance, and an L1 Loss on the normals as done in [37]. During inference, we also feed the predicted points and normals to our PSR solver and run Marching Cubes to obtain meshes.

**Metrics:** We consider Chamfer Distance, Normal Consistency and F-Score with the default threshold of $1\%$ for evaluation, and also report optimization & inference time.

### 4.1 Optimization-based 3D Reconstruction

In this part, we investigate whether our method can be used for the single-object surface reconstruction task from unoriented point clouds or scans. We consider three different types of 3D inputs: point clouds sampled from synthetic meshes [71] with Gaussian noise, real-world scans [65], and high-resolution raw scans of humans with comparably little noise [6].

Fig. 2 and Table 2 show that our method achieves superior performance compared to both classical methods and network-based approaches. Note that the objects considered in this task are challenging

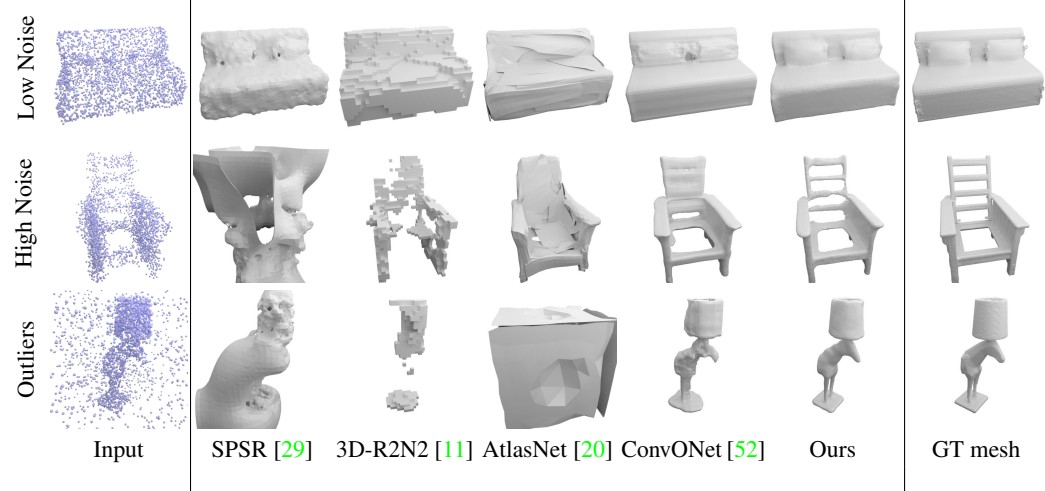

Figure 3: **3D Reconstruction from Point Clouds on ShapeNet.** Comparison of SAP to baselines on 3 different setups. More results can be found in supplementary.

| | (a) Noise=0.005 | | | (b) Noise=0.025 | | | (c) Noise=0.005, Outliers=50% | | | |
|---|---|---|---|---|---|---|---|---|---|---|
| | Chamfer-$L_1$ | F-Score | Normal C. | Chamfer-$L_1$ | F-Score | Normal C. | Chamfer-$L_1$ | F-Score | Normal C. | Runtime |
| SPSR [29] | 0.298 | 0.612 | 0.772 | 0.499 | 0.324 | 0.604 | 1.317 | 0.164 | 0.636 | - |
| PSGN [17] | 0.147 | 0.259 | - | 0.151 | 0.247 | - | 0.736 | 0.007 | - | 0.010 s |
| 3D-R2N2 [11] | 0.172 | 0.400 | 0.715 | 0.173 | 0.418 | 0.710 | 0.202 | 0.387 | 0.709 | 0.015 s |
| AtlasNet [20] | 0.093 | 0.708 | 0.855 | 0.117 | 0.527 | 0.821 | 1.822 | 0.057 | 0.609 | 0.025 s |
| ConvONet [52] | 0.044 | 0.942 | 0.938 | 0.066 | 0.849 | 0.913 | 0.052 | 0.916 | 0.929 | 0.327 s |
| Ours (w/o $\mathcal{L}_{\text{DPSR}}$) | 0.044 | 0.942 | 0.935 | 0.067 | 0.841 | 0.907 | 0.085 | 0.819 | 0.903 | 0.064 s |
| **Ours** | **0.034** | **0.975** | **0.944** | **0.054** | **0.896** | **0.917** | **0.038** | **0.959** | **0.936** | 0.064 s |

Table 3: **3D Reconstruction from Point Clouds on ShapeNet.** Quantitative comparison between our method and baselines on the ShapeNet dataset (mean over 13 classes).

due to their complex geometry, thin structures, noisy and incomplete observations. While some of the baseline methods fail completely on these challenging objects, our method achieves robust performance across all datasets.

In particular, Fig. 2 shows that IGR occasionally creates meshes in free space, as this is not penalized by its optimization objective when point clouds are unoriented. Both, Point2Mesh and our method alleviate this problem by optimizing for the Chamfer distance between the estimated mesh and the input point clouds. However, Point2Mesh requires an initial mesh as input of which the topology cannot be changed during optimization. Thus, it relies on SPSR to provide an initial mesh for objects with genus larger than 0 and suffers from inaccurate initialization [23]. Furthermore, compared to both IGR and Point2Mesh, our method converges faster.

While SPSR is even more efficient, it suffers from incorrect normal estimation on noisy input point clouds, which is a non-trivial task on its own. In contrast, our method demonstrates more robust behavior as we optimize points and normals guided by the Chamfer distance. Note that in this *single* object reconstruction task, our method is not able to complete large unobserved regions (e.g., the bottom of the person's feet in Fig. 2 is unobserved and hence not completed). This limitation can be addressed using learning-based object-level reconstruction as discussed next.

To analyze whether our proposed differentiable Poisson solver is also beneficial for learning-based reconstruction, we evaluate our method on the single object reconstruction task using noise and outlier-augmented point clouds from ShapeNet as input to our method. We investigate the performance for three different noise levels: (a) Gaussian noise with zero mean and standard deviation 0.005, (b) Gaussian noise with zero mean and standard deviation 0.025, (c) 50% points have the same noise as in a) and the other 50% points are outliers uniformly sampled inside the unit cube.

| | 128³ | | | | 256³ | | | | Chamfer | F-Score | NormalC |
|---|---|---|---|---|---|---|---|---|---|---|---|
| | Enc. | Grid | MC | Total | Enc. | Grid | MC | Total | | | |
| | | | | | | | | | Offset 1× | 0.041 | 0.952 | 0.928 |

| | 128³ | | | | 256³ | | | |
|---|---|---|---|---|---|---|---|---|
| | Enc. | Grid | MC | Total | Enc. | Grid | MC | Total |
| ConvONet | **0.010** | 0.280 | **0.037** | 0.327 | **0.010** | 3.798 | **0.299** | 4.107 |
| **Ours** | 0.013 | **0.012** | 0.039 | **0.064** | 0.019 | **0.140** | 0.374 | **0.533** |

| | Chamfer | F-Score | NormalC |
|---|---|---|---|
| Offset 1× | 0.041 | 0.952 | 0.928 |
| Offset 3× | 0.039 | 0.958 | 0.934 |
| Offset 5× | 0.039 | 0.957 | 0.934 |
| Offset 7× | **0.038** | **0.959** | **0.936** |
| 2D Enc. | 0.043 | 0.939 | 0.928 |
| 3D Enc. | **0.038** | **0.959** | **0.936** |

Table 4: **Ablation Study.** Left: Runtime breakdown (encoding, grid evaluation, marching cubes) for ConvONet vs. ours in seconds. Right: Ablation over number of offsets and 2D vs. 3D encoders.

## 4.2 Learning-based Reconstruction on ShapeNet

Fig. 3 and Table 3 show our results. Compared to the baselines, our method achieves similar or better results on all three metrics. The results show that, in comparison to directly using Chamfer loss on point positions and L1 loss on point normals, our DPSR loss can produce better reconstructions in all settings as it directly supervises the indicator grid which implicitly determines the surface through the Poisson equation. SPSR fails when the noise level is high or when there are outliers in the input point cloud. We achieve significantly better performances than other representations such as point clouds, meshes, voxel grids and patches. Moreover, we find that our method is robust to strong outliers. We refer to the supplementary for more detailed visualizations on how SAP handles outliers.

Table 3 also reports the runtime for setting (a) for all GPU-accelerated methods using a single NVIDIA GTX 1080Ti GPU, averaged over all objects of the ShapeNet test set. The baselines [11, 17, 20] demonstrate fast inference time but suffer in terms of reconstruction quality while the neural implicit model [52] attains high quality reconstructions but suffers from slow inference. In contrast, our method is able to produce competitive reconstruction results at reasonably fast inference time. In addition, since ConvONet and our method share a similar reconstruction pipeline, we provide a more detailed breakdown of the runtime at a resolution of $128^3$ and $256^3$ voxels in Table 4. We use the default setup from ConvONet[5]. As we can see from Table 4, the difference in terms of point encoding and Marching Cubes is marginal, but we gain more than $20\times$ speed-up over ConvONet in evaluating the indicator grid. In total, we are roughly $5\times$ and $8\times$ faster regarding the total inference time at a resolution of $128^3$ and $256^3$ voxels, respectively.

## 4.3 Ablation Study

In this section, we investigate different architecture choices in the context of learning-based reconstruction. We conduct our ablation experiments on ShapeNet for the third setup (most challenging).

**Number of Offsets:** From Table 4 we notice that predicting more offsets per input point leads to better performance. This can be explained by the fact that with more points near the object surface, geometric details can be better preserved.

**Point Cloud Encoder:** Here we compare two different point encoder architectures proposed in [52]: a 2D encoder using 3 canonical planes at a resolution of $64^2$ pixels and a 3D encoder using a feature volume with a resolution of $32^3$ voxels. We find that the 3D encoder works best in this setting and hypothesize that this is due to the representational alignment with the 3D indicator grid.

## 5 Conclusion

We introduce Shape-As-Points, a novel shape representation which is lightweight, interpretable and produces watertight meshes efficiently. We demonstrate its effectiveness for the task of surface reconstruction from unoriented point clouds in both optimization-based and learning-based settings. Our method is currently limited to small scenes due to the cubic memory requirements with respect to the indicator grid resolution. We believe that processing scenes in a sliding-window manner and space-adaptive data structures (e.g., octrees) will enable extending our method to larger scenes. Point cloud-based methods are broadly used in real-world applications ranging from household robots to self-driving cars, and hence share the same societal opportunities and risks as other learning-based 3D reconstruction techniques.

---

[5]To be consistent, we use the Marching Cubes implementation from [60] for both ConvONet and ours.

**Acknowledgement:** Andreas Geiger was supported by the ERC Starting Grant LEGO-3D (850533) and the DFG EXC number 2064/1 - project number 390727645. The authors thank the Max Planck ETH Center for Learning Systems (CLS) for supporting Songyou Peng and the International Max Planck Research School for Intelligent Systems (IMPRS-IS) for supporting Michael Niemeyer. This work was supported by an NVIDIA research gift. We thank Matthias Niessner, Thomas Funkhouser, Hugues Hopp, Yue Wang for helpful discussions in early stages of this project. We also thank Xu Chen, Christian Reiser, Rémi Pautrat for proofreading.

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
