# Supplementary Material for
# Shape As Points: A Differentiable Poisson Solver

**Songyou Peng**[1,2]     **Chiyu "Max" Jiang**[*][†]     **Yiyi Liao**[2,3][†]     **Michael Niemeyer**[2,3]

**Marc Pollefeys**[1,4]          **Andreas Geiger**[2,3]

[1]ETH Zurich          [2]Max Planck Institute for Intelligent Systems, Tübingen
[3]University of Tübingen          [4]Microsoft

In this **supplementary document**, we first provide derivation details for our Differentiable Poisson Solver in Section 1. In Section 2, we provide implementation details for our optimization-based and learning-based methods. Next, we supply discussions on the easy initialization property of our Shape-As-Points representation in Section 3. Additional results and ablations for the optimization-based and learning-based reconstruction can be found in Section 4 and Section 5, respectively.

## 1    Derivations for Differentiable Poisson Solver

### 1.1    Point Rasterization

Given the origin of the voxel grid $\mathbf{c}_0 = (x_0, y_0, z_0)$, and the size of each voxel $\mathbf{s} = (s_x, s_y, s_z)$, we scatter the point normal values to the voxel grid vertices, weighted by the trilinear interpolation weights. For a given point $\mathbf{p}_i := (\mathbf{c}_i, \mathbf{n}_i) \in \{\mathbf{p}_i, i = 1, 2, \cdots, N\}$, with point location $\mathbf{c}_i = (x_i, y_i, z_i)$ and point normal $\mathbf{n}_i = (\hat{x}_i, \hat{y}_i, \hat{z}_i)$, we can compute the neighbor indices as $\{\mathbf{j}\}$, where $\mathbf{j} = (j_x, j_y, j_z) \in (\left\lfloor \frac{x_i - x_0}{s_x} \right\rfloor, \left\lceil \frac{x_i - x_0}{s_x} \right\rceil) \times (\left\lfloor \frac{y_i - y_0}{s_y} \right\rfloor, \left\lceil \frac{y_i - y_0}{s_y} \right\rceil) \times (\left\lfloor \frac{z_i - z_0}{s_z} \right\rfloor, \left\lceil \frac{z_i - z_0}{s_z} \right\rceil)$. Here $\lfloor \rfloor$ and $\lceil \rceil$ denote the floor and ceil operators for rounding integers. We denote the trilinear sampling weight function as $\mathcal{T}(\mathbf{c}_p, \mathbf{c}_v, \mathbf{s})$, where $\mathbf{c}_p$ and $\mathbf{c}_v$ denote the location of the point and the grid vertex. The contribution from point $\mathbf{p}_i$ to voxel grid vertex $\mathbf{j}$ can be computed as:

$$\mathbf{v}_{j \leftarrow i} = \mathcal{T}(\mathbf{c}_i, \mathbf{s} \odot \mathbf{j} + \mathbf{c}_0, \mathbf{s})\mathbf{n}_i \tag{1}$$

Hence the value at grid index $\mathbf{j} \in r \times r \times r$ can be computed via summing over all neighborhood points:

$$\mathbf{v}_j = \sum_{i \in \mathcal{N}_j} \mathcal{T}(\mathbf{c}_i, \mathbf{s} \odot \mathbf{j} + \mathbf{c}_0, \mathbf{s})\mathbf{n}_i \tag{2}$$

where $\mathcal{N}_j$ denotes the set of point indices in the neighborhood of vertex $j$.

### 1.2    Spectral Methods for Solving PSR

We solve the PDEs using spectral methods [3]. In three dimensions, the multidimensional Fourier Transform and Inverse Fourier Transform are defined as:

$$\tilde{f}(\mathbf{u}) := \text{FFT}(f(\mathbf{x})) = \iiint_{\infty}^{\infty} f(\mathbf{x}) e^{-2\pi i \mathbf{x} \cdot \mathbf{u}} d\mathbf{x} \tag{3}$$

$$f(\mathbf{x}) := \text{IFFT}(\tilde{f}(\mathbf{u})) = \iiint_{\infty}^{\infty} \tilde{f}(\mathbf{u}) e^{2\pi i \mathbf{x} \cdot \mathbf{u}} d\mathbf{u} \tag{4}$$

where $\mathbf{x} := (x, y, z)$ are the spatial coordinates, and $\mathbf{u} := (u, v, w)$ represent the frequencies corresponding to $x, y$ and $z$. Derivatives in the spectral space can be analytically computed:

$$\frac{\partial}{\partial x_j} f(\mathbf{x}) = \iiint_{\infty}^{\infty} 2\pi i x_j \tilde{f}(\mathbf{u}) e^{2\pi i \mathbf{x} \cdot \mathbf{u}} d\mathbf{u} = \text{IFFT}(2\pi i x_j \tilde{f}(\mathbf{u}))$$

---

[*]Work done while at UC Berkeley.

[†]Corresponding authors.

35th Conference on Neural Information Processing Systems (NeurIPS 2021).

In discrete form, we have the rasterized point normals $\mathbf{v} := (v_x, v_y, v_z)$, where $v_x, v_y, v_z \in \mathbb{R}^n$. Hence in spectral domain, the divergence of the rasterized point normals can be written as:

$$\text{FFT}(\nabla \cdot \mathbf{v}) = 2\pi i(\mathbf{u} \cdot \tilde{\mathbf{v}}) \tag{5}$$

The Laplacian operator can be simply written as:

$$\text{FFT}(\nabla^2) = -4\pi^2 ||\mathbf{u}||^2 \tag{6}$$

Therefore, the unnormalized solution to the Poisson Equations $\tilde{\chi}$, not accounting for boundary conditions, can be written as:

$$\tilde{\chi} = \tilde{g}_{\sigma,r}(\mathbf{u})\frac{i\mathbf{u} \odot \tilde{\mathbf{v}}}{-2\pi||\mathbf{u}||^2} \qquad \tilde{g}_{\sigma,r}(\mathbf{u}) = \exp\left(-2\frac{\sigma^2||\mathbf{u}||^2}{r^2}\right) \tag{7}$$

Where $\tilde{g}_{\sigma,r}(\mathbf{u})$ is a Gaussian smoothing kernel of bandwidth $\sigma$ for grid resolution of $r$ in the spectral domain to mitigate the ringing effects as a result of the Gibbs phenomenon from rasterizing the point normals. Please refer to Section 4.3 for a more in-depth discussion to motivate the use of the smoothing parameter, as well as related ablation studies on our parameter choice for $\sigma$.

The unnormalized indicator function in the physical domain $\chi'$ can be obtained via inverse Fourier Transform:

$$\chi' = \text{IFFT}(\tilde{\chi}) \tag{8}$$

We further normalize the indicator field to incorporate the boundary condition that the indicator field is valued at zero at point locations and valued $\pm 0.5$ inside and outside the shapes.

$$\chi = \underbrace{\frac{m}{\text{abs}(\chi'|_{\mathbf{x}=0})}}_{\text{scale}} \underbrace{\left(\chi' - \frac{1}{|\{\mathbf{c}\}|}\sum_{\mathbf{c}\in\{\mathbf{c}\}}\chi'|_{\mathbf{x}=\mathbf{c}}\right)}_{\text{subtract by mean}} \tag{9}$$

## 2   Implementation Details

In this section, we provide implementation details for baselines and our method for both settings, optimization-based and the learning-based reconstruction.

**Optimization-based 3D reconstruction:**   We use the official implementation of IGR[3] [7] and Point2Mesh[4] [9]. We optimize IGR for 15000 iterations on each object until convergence. For Point2Mesh, we follow the official implementation and use 6000 iterations for each object. We generate the initial mesh required by Point2Mesh following the description of the original paper. Specifically, the initial mesh is provided as the convex hull of the input point cloud for objects with a genus of zero. If the genus is larger than zero, we apply the watertight manifold algorithm [10] using a low-resolution octree reconstruction on the output mesh of SPSR to obtain a coarse initial mesh.

For our method, we follow the coarse-to-fine and resampling strategy described in the main paper (Section 3.2). To smooth the output mesh as well as to stabilze the optimization process, we gradually increase the Gaussian smoothing parameter $\sigma$ in (7) when increasing the grid resolution: $\sigma = 2$ for a grid resolution of $32^3$ and $64^3$, $\sigma = 3$ when the grid resolution is $128^3$. At the final resolution of $256^3$, we use $\sigma = 3$ for objects with more details (e.g. objects in SRB [18] and D-FAUST [2], and $\sigma = 5$ for the input points with noises (Thingi10K [20]). We use the Adam optimizer [13] with a learning rate decay. The learning rate is set to $2 \times 10^{-3}$ at the initial resolution of $32^3$ with a decay of $0.7$ after every increase of the grid resolution. Moreover, we run 1000 iterations at every grid resolution of $32^3$, $64^3$ and $128^3$, and 200 iterations for $256^3$. 20000 source points and normals are used by our method to represent the final shapes for all objects.

---

[3]https://github.com/amosgropp/IGR
[4]https://github.com/ranahanocka/point2mesh

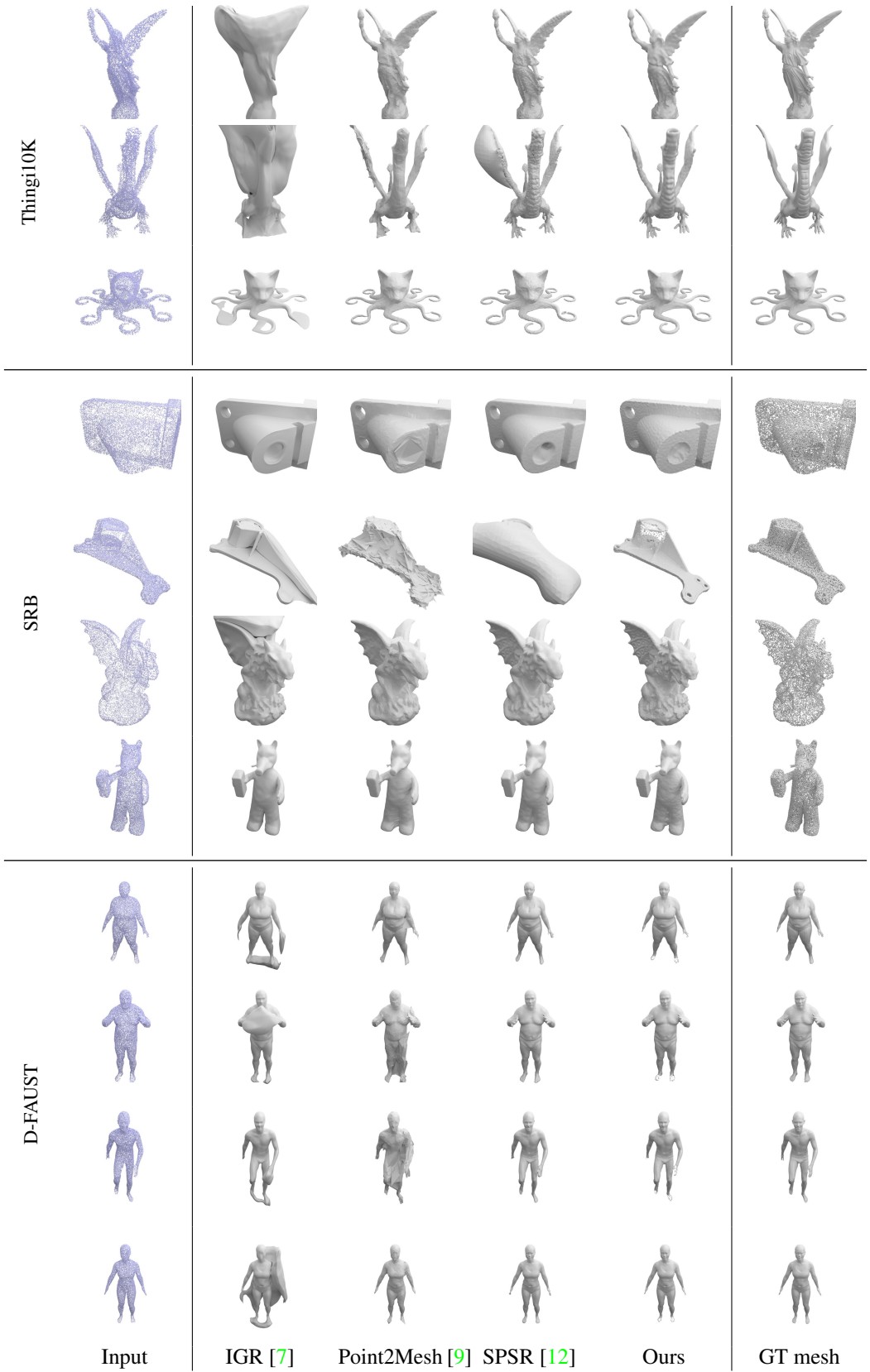

Figure 1: **Optimization-based 3D Reconstruction.** Here we show all the other 11 out of 15 objects used for comparison (We have already shown 4 objects in Fig. 2 in main paper). Input point clouds are downsampled for visualization. Note that the ground truth of SRB is provided as point clouds.

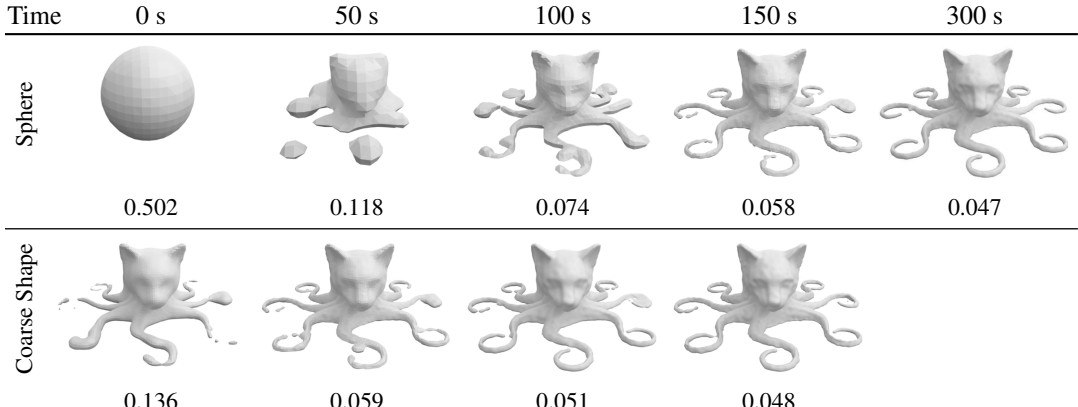

Figure 2: **Different Geometric Initialization under Optimization Setting**. We compare the reconstructions of SAP initialized from a sphere and the coarse geometry. The number below each image indicates the Chamfer Distance to GT mesh.

| Iterations | 10K | 50K | 100K | 200K | Best |
|---|---|---|---|---|---|
| ConvONet [16] | 0.082 | 0.058 | 0.055 | 0.050 | 0.044 |
| **Ours** | **0.041** | **0.036** | **0.035** | **0.034** | **0.034** |

Table 1: **Training Progress**. We show the Chamfer distance at different training iterations evaluated in the Shapenet test set with 3K input points ((noise level=0.005). Our method uses geometric initialization and converges much faster than ConvONet.

**Learning-based 3D reconstruction:** For AtlasNet [8], we use the official implementation[5] with 25 parameterizations. We change the number of input points from 2500 (default) to 3000 for our setting. Depending on the experiment, we adde different noise levels or outlier points (see Section 4.2 in main paper). We train ConvONet [16], PSGN [6], and 3D-R2N2 [4] for at least 300000 iterations, and use Adam optimizer [13] with a learning rate of $10^{-4}$ for all methods.

We train our method as well as Ours (w/o $\mathcal{L}_{\text{DPSR}}$) for all 3 noise levels for 300000 iterations (roughly 2 days with 2 GTX 1080Ti GPUs) and use Adam optimizer with a learning rate of $5 \times 10^{-4}$. We consider a batch size of 32. To generate the ground truth PSR indicator field $\chi$ in Eq. (9) of the main paper, first we sample 100000 points and the corresponding point normals from the ground truth mesh, and input to our DPSR at a grid resolution of $128^3$.

## 3 Discussions on "Easy Initialization" Property of SAP

As mentioned in the main paper, it is easy to initialize SAP with a given geometry such as template shapes or noisy observations. Here we provide further discussions with some experiments under both optimization and learning settings of SAP.

From all the results that we have shown so far under the optimization setting, we chose to start from a sphere, since we intended to demonstrate that if our method is able to produce decent 3D reconstruction even starting from a sphere, we can also faithfully reconstruct from a coarse or noisy shape since it is a simpler task, and it should converge faster. In Fig. 2, we show a comparison between initialization from a sphere and coarse shape. As can be observed, when starting from points and normals sampled from a coarse shape, our method indeed converges faster. In terms of accuracy, both results are equivalent, i.e., there is no better local minima attained by the optimization process.

In the learning setting, what we want to emphasize is *geometric initialization* instead of network initialization. For neural-implicit methods, weight initializations can only be analytically derived for simple shapes like spheres [1]. In contrast, since our networks only need to predict the offset and normal fields for the input point cloud, this input point cloud is directly treated as the geometric

---
[5]https://github.com/ThibaultGROUEIX/AtlasNet

| Dataset | Method | Chamfer-$L_1$ ($\downarrow$) | F-Score ($\uparrow$) | Normal C. ($\uparrow$) |
|---|---|---|---|---|
| Thingi10K | Ours (w/o resampling) | 0.061 | 0.897 | 0.902 |
| | Ours | **0.053** | **0.941** | **0.947** |
| DGP | Ours (w/o resampling) | 0.077 | 0.813 | – |
| | Ours | **0.067** | **0.848** | – |
| D-FAUST | Ours (w/o resampling) | 0.044 | 0.964 | 0.952 |
| | Ours | **0.043** | **0.965** | **0.959** |

Table 2: **Ablation Study of Resampling Strategy**. On all datasets, our resampling strategy leads to improved results. For D-FAUST, the increase is the lowest because the supervision point clouds are noise free. Note that normal consistency cannot be evaluated on SRB as this dataset provides only unoriented point clouds.

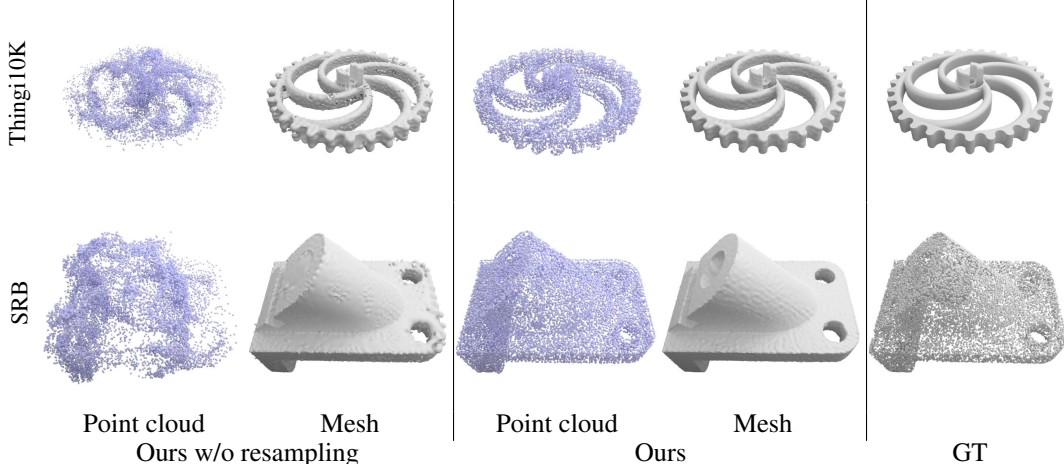

Figure 3: **Ablation Study of Resampling Strategy**. We show the optimized point cloud and the reconstructed mesh without and with the resampling strategy. Using the point resampling strategy leads to a more uniformly distributed point cloud and better shape reconstruction.

initialization. In Table 1, we show the Chamfer distance at different training iterations evaluated in the ShapeNet test set with 3K input points (noise level=0.005). As can be seen, our method enables much faster training convergence than the neural-implicit method ConvONet [16], which illustrates the effectiveness of the geometric initialization directly from input point clouds.

# 4 Optimization-based 3D Reconstruction

## 4.1 Qualitative Comparison of All Objects in 3 Datasets

As a complementary to Fig. 2 in the main paper, Fig. 1 shows qualitative comparisons of the remaining 11 objects in the optimization-based setting, including 3 objects in Thingi10K [20], 4 in SRB [18] and 4 in D-FAUST [2].

## 4.2 Ablation Study of Point Resampling Strategy

In Table 2 and Fig. 3, we compare the reconstructed shapes with and without the proposed resampling strategy. Our method is able to produce reasonable reconstructions even without the resampling strategy, but the shapes are much noisier. Since we directly optimize the source point positions and normals without any additional constraints, the optimized point clouds can be unevenly distributed as shown in Fig. 3. This limits the representational expressivity of the point clouds given the same number of points. The resampling strategy acts as a regularization to enforce a uniformly distributed point cloud, which leads to better surface reconstruction.

### 4.3 Analysis on the Gaussian Smoothing Parameter $\sigma$

**Why we need a Gaussian:** The Gaussian serves as a regularizer to the smoothness of the solved implicit function. Not using a Gaussian is equivalent to using a Gaussian kernel with $\sigma = 0$. Fig. 4 below motivates the use of our sigma parameter.

**Why use a Gaussian in the spectral domain:** First, the FFT of a Gaussian remains a Gaussian. Second, convolution of a Gaussian in the physical domain is equivalent to a dot product with a Gaussian in the spectral domain and a dot product in the spectral domain is more efficient than convolution in the physical domain: $\mathcal{O}(N \log N)$ vs $\mathcal{O}(N^2)$, where $n$ is the resolution of a regular grid and $N = n^3$.

**Ablation study for the Gaussian smoothing parameter $\sigma$:** We study the effect of the Gaussian smoothing parameter $\sigma$ at a resolution of $256^3$. As visualized in Fig. 4, we can obtain faithful reconstructions given different $\sigma$ values. Nevertheless, we can notice that lower $\sigma$ can preserve details better but also is prone to noise, while high $\sigma$ results in smooth shapes but can also lead to losing of details. In practice, $\sigma$ can be chosen according to the noise level of the target point cloud. In the results depicted in Fig. 1 and Table 2 in main paper, we choose $\sigma = 3$ for SRB and D-FAUST dataset and $\sigma = 5$ for Thingi10K dataset.

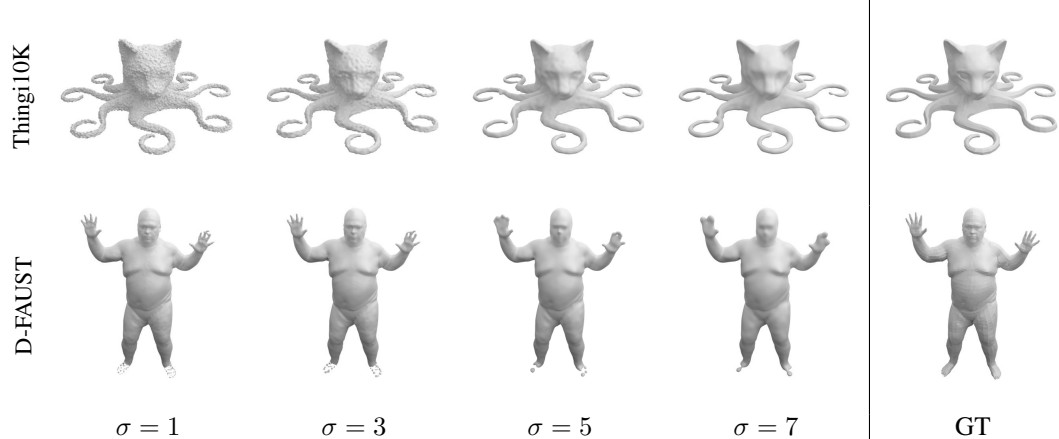

Figure 4: **Ablation Study of the Gaussian Smoothing Parameter $\sigma$.** Low $\sigma$ preserves details better but is prone to noise, while high $\sigma$ results in smooth shapes but also lead to detail losing.

### 4.4 Preliminary Results on Multi-view Reconstruction with Differentiable Rendering

We have validated the effectiveness of our Shape-As-Points representation with experiments on 3D reconstruction from 3D point clouds. To further show the flexibility of SAP, here we also provide a proof-of-concept experiments for 3D reconstruction from multi-view images. Indeed, our differentiable point-to-mesh layer under the optimization-based setting can be naturally integrated with a differentiable renderer.

We proceed as follows. After obtaining the mesh in each iteration, for each 2D image, we rasterize the 3D mesh using `rasterize_meshes` in PyTorch3D [17] to find the corresponding surface points for each pixel, and use an MLP to predict their RGB colors. An $L_2$ loss is applied to the predicted and input RGB images. In addition, we also apply a silhouette loss using a differentiable silhouette renderer in PyTorch3D.

In Fig. 5 we show some preliminary results, where we test our implementation on 1 synthetic object (cow) with 20 images and 2 real-world objects in DTU dataset [11] with around 50 images. As can be observed, our method is indeed able to reconstruct detailed geometry using only 2D supervisions, and we consider this as an exciting extension for future work.

We further remark that recent methods based on neural implicit representations [14, 15, 19] can attain high-quality 3D reconstruction, but their optimization process is slow due to dense evaluation in the

Synthetic                                    DTU [11]

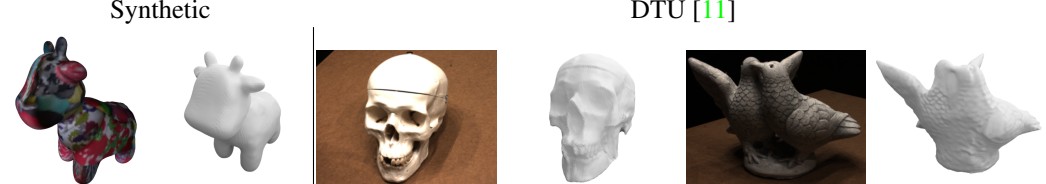

Figure 5: **SAP Multi-view Reconstruction with Differentiable Rendering.** We show the reconstruction results on a synthetic dataset and a real-world dataset [11].

ray marching step. In contrast, in each iteration our method outputs an explicit mesh. Therefore, no expensive ray marching is required to find the surface, leading to faster inference.

# 5    Learning-based 3D Reconstruction

## 5.1    Visualization of How SAP Handles Noise and Outliers

In this section, we visualize how our trained models handle noise and outliers during inference.

**Noise Handling:**  We can see from the top row of Fig. 6 that, compared to the input point cloud, the updated SAP points are densified because we predict $k = 7$ offsets per input point. More importantly, all SAP points are located roughly on the surface, which leads to enhanced reconstruction quality.

**Outlier Handling:**  We also visualize how SAP handles outlier points at the bottom row of Fig. 6. The arrows' length represents the magnitude of the predicted normals. There are two interesting observations: a) A large amount of outlier points in the input are moved near to the surface. b) Some outlier points still remain outliers. For these points, the network learns to predict normals with a very small magnitude/norm as shown in the zoom-in view (we do not normalize the point normals to unit length). In this way, those outlier points are "muted" when being passed to the DPSR layer such that they do not contribute to the final reconstruction.

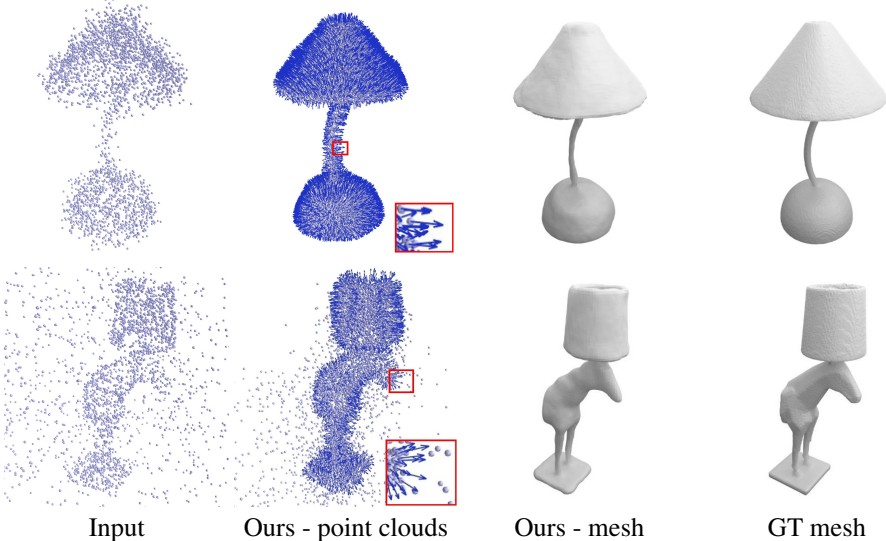

Input              Ours - point clouds        Ours - mesh              GT mesh

Figure 6: **Visualization of SAP Handling Noise and Outliers.** The length of arrows represents the magnitude of normals. SAP point clouds are downsampled for better visualization.

## 5.2    Additional Results for Reconstruction from Noisy Point Clouds

In Table 3 we provide quantitative results on all 13 classes of the ShapeNet subset of Choy et al. [4]. In Fig. 7, we show additional qualitative comparison on ShapeNet.

**(a) Noise = 0.005**

| category | Chamfer-$L_1$ SPSR | PSGN | 3D-R2N2 | AtlasNet | ConvONet | Ours (w/o $\mathcal{L}_{DPSR}$) | Ours | F-Score SPSR | PSGN | 3D-R2N2 | AtlasNet | ConvONet | Ours (w/o $\mathcal{L}_{DPSR}$) | Ours | NC SPSR | PSGN | 3D-R2N2 | AtlasNet | ConvONet | Ours (w/o $\mathcal{L}_{DPSR}$) | Ours |
|---|---|---|---|---|---|---|---|---|---|---|---|---|---|---|---|---|---|---|---|---|---|
| airplane | 0.437 | 0.102 | 0.151 | 0.064 | 0.034 | 0.040 | **0.027** | 0.551 | 0.476 | 0.382 | 0.827 | 0.965 | 0.940 | **0.981** | 0.747 | - | 0.669 | 0.854 | **0.931** | 0.919 | **0.931** |
| bench | 0.544 | 0.129 | 0.153 | 0.073 | 0.035 | 0.041 | **0.032** | 0.430 | 0.266 | 0.431 | 0.786 | 0.965 | 0.949 | **0.979** | 0.649 | - | 0.691 | 0.820 | **0.921** | 0.915 | 0.920 |
| cabinet | 0.154 | 0.164 | 0.167 | 0.112 | 0.047 | 0.044 | **0.037** | 0.728 | 0.137 | 0.412 | 0.603 | 0.955 | 0.952 | **0.975** | 0.835 | - | 0.786 | 0.875 | 0.956 | 0.951 | **0.957** |
| car | 0.180 | 0.132 | 0.197 | 0.099 | 0.075 | 0.061 | **0.045** | 0.729 | 0.211 | 0.348 | 0.642 | 0.849 | 0.875 | **0.928** | 0.783 | - | 0.719 | 0.827 | 0.893 | 0.886 | **0.897** |
| chair | 0.369 | 0.168 | 0.181 | 0.114 | 0.046 | 0.047 | **0.036** | 0.473 | 0.152 | 0.393 | 0.629 | 0.939 | 0.939 | **0.979** | 0.715 | - | 0.673 | 0.829 | 0.943 | 0.940 | **0.952** |
| display | 0.280 | 0.160 | 0.170 | 0.089 | 0.036 | 0.036 | **0.030** | 0.544 | 0.175 | 0.401 | 0.727 | 0.971 | 0.975 | **0.990** | 0.749 | - | 0.747 | 0.905 | 0.968 | 0.967 | **0.972** |
| lamp | 0.278 | 0.207 | 0.243 | 0.137 | 0.059 | 0.069 | **0.047** | 0.586 | 0.204 | 0.333 | 0.562 | 0.892 | 0.897 | **0.959** | 0.765 | - | 0.598 | 0.759 | 0.900 | 0.899 | **0.921** |
| loudspeaker | 0.148 | 0.205 | 0.199 | 0.142 | 0.063 | 0.058 | **0.041** | 0.731 | 0.107 | 0.405 | 0.516 | 0.892 | 0.900 | **0.957** | 0.843 | - | 0.735 | 0.867 | 0.938 | 0.935 | **0.950** |
| rifle | 0.409 | 0.091 | 0.147 | 0.051 | 0.028 | 0.027 | **0.023** | 0.590 | 0.615 | 0.381 | 0.877 | 0.980 | 0.982 | **0.990** | 0.788 | - | 0.700 | 0.837 | 0.929 | 0.935 | **0.937** |
| sofa | 0.227 | 0.144 | 0.160 | 0.091 | 0.041 | 0.039 | **0.032** | 0.712 | 0.184 | 0.427 | 0.717 | 0.953 | 0.960 | **0.982** | 0.826 | - | 0.754 | 0.888 | 0.958 | 0.957 | **0.963** |
| table | 0.393 | 0.166 | 0.177 | 0.102 | 0.038 | 0.043 | **0.033** | 0.442 | 0.158 | 0.404 | 0.692 | 0.967 | 0.958 | **0.986** | 0.706 | - | 0.734 | 0.867 | 0.959 | 0.954 | **0.962** |
| telephone | 0.281 | 0.110 | 0.130 | 0.054 | 0.027 | 0.026 | **0.023** | 0.674 | 0.317 | 0.484 | 0.867 | 0.989 | 0.992 | **0.997** | 0.805 | - | 0.847 | 0.957 | 0.983 | 0.982 | **0.984** |
| vessel | 0.181 | 0.130 | 0.169 | 0.078 | 0.043 | 0.043 | **0.030** | 0.771 | 0.363 | 0.394 | 0.757 | 0.931 | 0.930 | **0.974** | 0.820 | - | 0.641 | 0.837 | 0.918 | 0.917 | **0.930** |
| mean | 0.299 | 0.147 | 0.173 | 0.093 | 0.044 | 0.044 | **0.034** | 0.612 | 0.259 | 0.400 | 0.708 | 0.942 | 0.942 | **0.975** | 0.772 | - | 0.715 | 0.855 | 0.938 | 0.935 | **0.944** |

**(b) Noise = 0.025**

| category | Chamfer-$L_1$ SPSR | PSGN | 3D-R2N2 | AtlasNet | ConvONet | Ours (w/o $\mathcal{L}_{DPSR}$) | Ours | F-Score SPSR | PSGN | 3D-R2N2 | AtlasNet | ConvONet | Ours (w/o $\mathcal{L}_{DPSR}$) | Ours | NC SPSR | PSGN | 3D-R2N2 | AtlasNet | ConvONet | Ours (w/o $\mathcal{L}_{DPSR}$) | Ours |
|---|---|---|---|---|---|---|---|---|---|---|---|---|---|---|---|---|---|---|---|---|---|
| airplane | 0.716 | 0.107 | 0.147 | 0.103 | 0.052 | 0.059 | **0.045** | 0.268 | 0.457 | 0.413 | 0.558 | 0.883 | 0.857 | **0.915** | 0.550 | - | 0.665 | 0.787 | 0.904 | 0.897 | **0.905** |
| bench | 0.661 | 0.133 | 0.154 | 0.101 | 0.056 | 0.060 | **0.050** | 0.296 | 0.255 | 0.446 | 0.587 | 0.872 | 0.862 | **0.920** | 0.551 | - | 0.683 | 0.797 | **0.887** | 0.879 | 0.885 |
| cabinet | 0.323 | 0.166 | 0.165 | 0.118 | 0.065 | 0.067 | **0.051** | 0.383 | 0.138 | 0.435 | 0.554 | 0.883 | 0.863 | **0.920** | 0.671 | - | 0.784 | 0.855 | 0.937 | 0.927 | **0.938** |
| car | 0.338 | 0.137 | 0.188 | 0.115 | 0.104 | 0.091 | **0.071** | 0.415 | 0.200 | 0.388 | 0.528 | 0.739 | 0.749 | **0.817** | 0.632 | - | 0.714 | 0.792 | **0.875** | 0.862 | 0.871 |
| chair | 0.524 | 0.176 | 0.191 | 0.126 | 0.071 | 0.073 | **0.058** | 0.263 | 0.141 | 0.387 | 0.527 | 0.818 | 0.799 | **0.882** | 0.585 | - | 0.666 | 0.811 | 0.915 | 0.905 | **0.920** |
| display | 0.409 | 0.166 | 0.167 | 0.111 | 0.057 | 0.057 | **0.047** | 0.321 | 0.164 | 0.431 | 0.554 | 0.889 | 0.881 | **0.925** | 0.600 | - | 0.743 | 0.884 | 0.946 | 0.944 | **0.951** |
| lamp | 0.457 | 0.210 | 0.261 | 0.146 | 0.090 | 0.101 | **0.076** | 0.319 | 0.195 | 0.329 | 0.455 | 0.754 | 0.734 | **0.841** | 0.617 | - | 0.588 | 0.737 | 0.866 | 0.859 | **0.881** |
| loudspeaker | 0.320 | 0.205 | 0.203 | 0.144 | 0.090 | 0.092 | **0.065** | 0.369 | 0.109 | 0.407 | 0.471 | 0.793 | 0.778 | **0.853** | 0.675 | - | 0.734 | 0.850 | 0.920 | 0.910 | **0.925** |
| rifle | 0.848 | 0.097 | 0.144 | 0.119 | 0.047 | 0.044 | **0.042** | 0.218 | 0.575 | 0.403 | 0.439 | 0.905 | 0.917 | **0.928** | 0.541 | - | 0.691 | 0.746 | 0.888 | 0.895 | **0.896** |
| sofa | 0.452 | 0.152 | 0.153 | 0.109 | 0.065 | 0.061 | **0.051** | 0.337 | 0.166 | 0.457 | 0.572 | 0.857 | 0.865 | **0.908** | 0.631 | - | 0.744 | 0.860 | 0.936 | 0.934 | **0.941** |
| table | 0.514 | 0.169 | 0.177 | 0.115 | 0.057 | 0.061 | **0.049** | 0.293 | 0.158 | 0.431 | 0.564 | 0.885 | 0.869 | **0.923** | 0.597 | - | 0.729 | 0.847 | 0.936 | 0.929 | **0.940** |
| telephone | 0.521 | 0.112 | 0.128 | 0.105 | 0.038 | 0.038 | **0.033** | 0.329 | 0.311 | 0.508 | 0.520 | 0.959 | 0.964 | **0.976** | 0.591 | - | 0.847 | 0.917 | 0.975 | 0.975 | **0.976** |
| vessel | 0.403 | 0.135 | 0.173 | 0.111 | 0.073 | 0.072 | **0.058** | 0.399 | 0.341 | 0.397 | 0.527 | 0.796 | 0.794 | **0.856** | 0.612 | - | 0.639 | 0.788 | 0.881 | 0.875 | **0.886** |
| mean | 0.499 | 0.151 | 0.173 | 0.117 | 0.066 | 0.067 | **0.054** | 0.324 | 0.247 | 0.418 | 0.527 | 0.849 | 0.841 | **0.896** | 0.604 | - | 0.710 | 0.821 | 0.913 | 0.907 | **0.917** |

**(c) Noise = 0.005, Outliers = 50%**

| category | Chamfer-$L_1$ SPSR | PSGN | 3D-R2N2 | AtlasNet | ConvONet | Ours (w/o $\mathcal{L}_{DPSR}$) | Ours | F-Score SPSR | PSGN | 3D-R2N2 | AtlasNet | ConvONet | Ours (w/o $\mathcal{L}_{DPSR}$) | Ours | NC SPSR | PSGN | 3D-R2N2 | AtlasNet | ConvONet | Ours (w/o $\mathcal{L}_{DPSR}$) | Ours |
|---|---|---|---|---|---|---|---|---|---|---|---|---|---|---|---|---|---|---|---|---|---|
| airplane | 1.573 | 0.745 | 0.164 | 2.113 | 0.041 | 0.213 | **0.031** | 0.093 | 0.011 | 0.405 | 0.027 | 0.938 | 0.667 | **0.970** | 0.621 | - | 0.650 | 0.561 | 0.920 | 0.864 | **0.923** |
| bench | 1.499 | 0.573 | 0.166 | 1.856 | 0.041 | 0.076 | **0.036** | 0.126 | 0.007 | 0.431 | 0.053 | 0.945 | 0.829 | **0.965** | 0.575 | - | 0.695 | 0.630 | **0.910** | 0.876 | 0.909 |
| cabinet | 1.060 | 0.712 | 0.175 | 1.472 | 0.052 | 0.065 | **0.042** | 0.248 | 0.004 | 0.399 | 0.083 | 0.938 | 0.868 | **0.950** | 0.659 | - | 0.778 | 0.639 | **0.950** | 0.925 | 0.950 |
| car | 1.262 | 0.536 | 0.200 | 1.844 | 0.087 | 0.092 | **0.057** | 0.177 | 0.009 | 0.351 | 0.059 | 0.812 | 0.764 | **0.893** | 0.634 | - | 0.711 | 0.620 | 0.884 | 0.863 | **0.888** |
| chair | 0.984 | 0.689 | 0.228 | 1.478 | 0.055 | 0.087 | **0.041** | 0.186 | 0.005 | 0.362 | 0.076 | 0.903 | 0.783 | **0.959** | 0.628 | - | 0.672 | 0.644 | 0.930 | 0.898 | **0.941** |
| display | 1.312 | 0.965 | 0.201 | 1.685 | 0.041 | 0.060 | **0.034** | 0.188 | 0.004 | 0.374 | 0.062 | 0.956 | 0.878 | **0.980** | 0.627 | - | 0.747 | 0.593 | 0.962 | 0.945 | **0.967** |
| lamp | 1.402 | 0.958 | 0.399 | 2.080 | 0.073 | 0.110 | **0.047** | 0.123 | 0.004 | 0.283 | 0.037 | 0.838 | 0.699 | **0.941** | 0.630 | - | 0.587 | 0.569 | 0.885 | 0.844 | **0.910** |
| loudspeaker | 0.930 | 0.905 | 0.224 | 1.392 | 0.037 | 0.093 | **0.050** | 0.264 | 0.003 | 0.376 | 0.093 | 0.861 | 0.792 | **0.927** | 0.673 | - | 0.679 | 0.519 | 0.916 | 0.907 | **0.929** |
| rifle | 1.689 | 0.479 | 0.163 | 2.442 | 0.037 | 0.055 | **0.026** | 0.066 | 0.022 | 0.386 | 0.012 | 0.953 | 0.888 | **0.985** | 0.627 | - | 0.700 | 0.666 | 0.950 | 0.933 | **0.956** |
| sofa | 1.267 | 0.607 | 0.172 | 1.656 | 0.047 | 0.064 | **0.037** | 0.211 | 0.006 | 0.412 | 0.080 | 0.934 | 0.866 | **0.967** | 0.655 | - | 0.756 | 0.666 | 0.950 | 0.933 | **0.956** |
| table | 1.159 | 0.913 | 0.202 | 1.581 | 0.044 | 0.082 | **0.037** | 0.166 | 0.004 | 0.405 | 0.081 | 0.950 | 0.810 | **0.972** | 0.618 | - | 0.737 | 0.672 | 0.951 | 0.915 | **0.954** |
| telephone | 1.458 | 0.851 | 0.146 | 1.890 | 0.030 | 0.036 | **0.025** | 0.173 | 0.005 | 0.461 | 0.047 | 0.983 | 0.971 | **0.994** | 0.668 | - | 0.839 | 0.589 | 0.980 | 0.975 | **0.982** |
| vessel | 1.530 | 0.639 | 0.189 | 2.200 | 0.054 | 0.072 | **0.036** | 0.108 | 0.009 | 0.387 | 0.032 | 0.890 | 0.814 | **0.956** | 0.652 | - | 0.635 | 0.568 | 0.907 | 0.886 | **0.921** |
| mean | 1.317 | 0.736 | 0.202 | 1.822 | 0.052 | 0.085 | **0.038** | 0.164 | 0.007 | 0.387 | 0.057 | 0.916 | 0.818 | **0.959** | 0.636 | - | 0.709 | 0.609 | 0.929 | 0.903 | **0.936** |

Table 3: **3D Reconstruction from Point Clouds on ShapeNet.** This table shows a per-category comparison of baselines and different variants of our approach. We train all methods on all 13 classes.

## 5.3 Comparison to Points2Surf [5]

Here we also provide a quantitative comparison to Points2Surf [5], which is also a learning-based 3D reconstruction method with noisy point clouds as input. We train and test on the lamp object class in ShapeNet with a noise level of 0.005. The reason why we do not train on the entire ShapeNet dataset and not testing on all 3 settings is the very long training and inference time of Points2Surf [5]. Due to the constraints of time and resources, we choose to compare only on 'lamp', a challenging class in ShapeNet, which has roughly 2000 objects for training and 450 objects for testing.

We use the official implementation[6] and 8 GTX 1080Ti GPUs, and it takes over 50 hours to reach 150 epochs (the convergence time as reported in the paper). In contrast, we train our method for only 30K iterations, which takes 2 GTX 1080Ti GPUs for less than 4 hours. Our results from 30K are close to the best model from 300K iterations. As shown in the Table 4, our method outperforms Points2Surf [5] in all metrics, and the inference speed is 3 orders of magnitude faster.

---

[6]https://github.com/ErlerPhilipp/points2surf

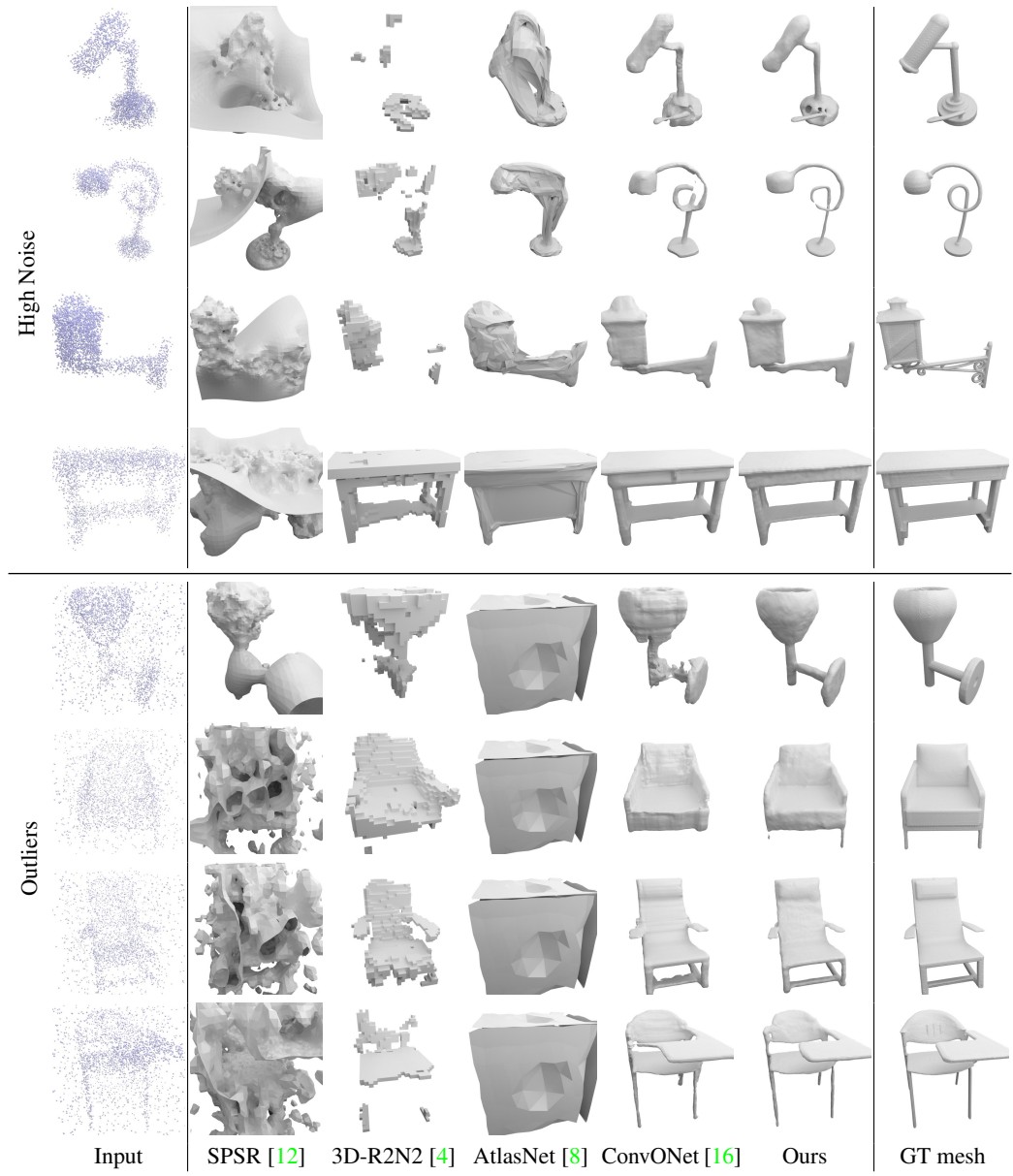

|  | Input | SPSR [12] | 3D-R2N2 [4] | AtlasNet [8] | ConvONet [16] | Ours | GT mesh |

Figure 7: **3D Reconstruction from Point Clouds on ShapeNet.** Comparison of SAP to baselines on 2 different setups. Compared to SPSR, our method can robustly estimate normals under noise and selectively ignore outliers. Compared to 3D-R2N2 and AtlasNet, SAP has greater representational expressivity that allows it to reconstruct fine geometric details. Compared to ConvONet, SAP produces a higher quality reconstruction at a fraction of its inference time.

|  | Chamfer-$L_1$ | F-Score | Normal C. | Inference time |
|---|---|---|---|---|
| Points2Surf [5] | 0.078 | 0.838 | 0.844 | 70 s |
| **Ours** | **0.048** | **0.950** | **0.918** | **0.064 s** |

Table 4: **Quantitative Comparison to Points2Surf [5].** We train and test on the ShapeNet 'lamp' class. Our model is trained only for 30K iterations but outperforms Points2Surf in all metrics with much shorter inference time.