# OpenReview forum: "Shape As Points: A Differentiable Poisson Solver"
_NeurIPS.cc/2021/Conference — NeurIPS 2021 Oral_

### Official Review · Reviewer_KwTG · 2021-07-13

**Rating:** 8
**Confidence:** 4

**Summary:**

The paper formulates a differentiable Poisson Surface Reconstruction layer, which maps an oriented point cloud to an indicator function over the interior of the object. The Poisson solver is done in the spectral domain on the GPU for computational efficiency. It shares many of its advantages with other neural implicit representations, but because it produces an indicator function over the whole space, there is no need to query a network over an entire voxel grid.

The layer is incorporated into two tasks: an optimization-based 3D shape reconstruction that fits a spherical point cloud initialization to a target point cloud, and a learning-based 3D reconstruction task that matches noisy point cloud input to a ground truth mesh. The non-differentiable step of Marching Cubes in these pipelines has its gradient approximated with the inverse surface normal, which is quite natural. The experimental results seem good, and beat out the existing state-of-the-art, and an ablation study is performed as well on the learning-based task to investigate the form of the feature extraction network there.

**Limitations And Societal Impact:**

I think the authors did a fair job of addressing these points.

**Main Review:**

I think the method is an interesting and novel implementation of the PSR algorithm, used to produce a better implicit representation than the existing state-of-the-art methods. I'm generally convinced by the experimental results, though I would appreciate some more information on the structure of the offset and normal prediction network. It was not quite clear to me why the particular architecture choices were made and whether other options were considered. The ablation study sticks to the convolutional point cloud encoder that they noted earlier, and considers two different versions, but does not consider other network architectures.

With respect to the layer, I was also curious to know about how much spectral resolution was used in the Poisson solver, and how the sigma was chosen for the Gaussian smoothing kernel.

**Time Spent Reviewing:**

2

---

> ### Author Response · Authors · 2021-08-09
> **Response to Reviewer KwTG**
>
> We thank the reviewer for considering our method interesting and novel. We address the remaining concerns below.
>
> **Explain why the particular network architecture is chosen and not consider other architectures.**
>
> We choose to build upon the architecture of ConvONet, which is a state-of-the-art method for surface reconstruction. In particular, ConvONet has demonstrated that its convolutional point cloud encoder outperforms previous point cloud encoders (e.g., PointNet) by a large margin. Moreover, we aim to study and benchmark SAP against neural implicit representations which have captured increasing interest in the 3D learning community, and ConvONet is a representative state-of-the-art method. To ensure a fair comparison to ConvONet we adopt their encoder architecture for our purpose and show improvements both quantitatively and qualitatively.
>
> **Some more information on the structure of the offset and normal prediction network**
>
> Thanks for the comment. For both offset and normal prediction networks, We use the same decoder architecture proposed in ConvONet: a 5-layer Multi-Layer Perceptron (MLP) with feature dimension 32 in each hidden layer. These two networks do not share weights. We will add this to the implementation details in the final version.
>
> **For PSR layer, how much spectral resolution was used in the Poisson solver**
>
> The spectral resolution is aligned with the grid resolution due to the fact that the FFT algorithm uses a uniformly sampled stencil. For the optimization-based setting, we start from a resolution of $32^3$ and gradually increase to $256^3$ (Line 176-179). For the learning-based setting, we consider resolutions of $128^3$ and $256^3$ (Table 4 and Line 281).
>
> **how the sigma was chosen for the Gaussian smoothing kernel**
>
> As discussed in Section 4.3 in the supplementary, a low sigma results in better details but might cause artifacts. In contrast, higher sigma values are more robust to the noise but lead to oversmoothing. For the optimization-based setting, we use a low sigma value for inputs with little noise (SRB, D-FAUST), and a higher sigma value for noisy inputs (Thingi10K). More details can be found in L59-62 in the supplementary. For learning-based 3D reconstruction, we use sigma=2 (low) for all experiments.

---

> > ### Comment · Reviewer_KwTG · 2021-08-26
> > **Post rebuttal comments**
> >
> > Thank you for the clarifications, and for agreeing to add some of these details and explanations to the text. My score remains the same, and I strongly support acceptance.

---

### Official Review · Reviewer_eo1g · 2021-07-14

**Rating:** 8
**Confidence:** 4

**Summary:**

This work introduces a differentiable points (+normals) to mesh layer based on a differentiable formulation of Poisson surface reconstruction.
First, the proposed method is validated in point cloud optimisation tasks where the objective function is expressed w.r.t surface meshes.
Then, the method is proposed as a way to represent 3D shapes with oriented point cloud in a learning setting.
Crucially, this results in a shape parameterization, dubbed Shape As Points (SAP), that is more lightweight and faster than neural implicit fields, while still yielding watertight shapes and being able to handle arbitrary topology.


**Limitations And Societal Impact:**

As stated above, I think the cubic memory requirement at training time should have been better pointed out in the paper.

**Main Review:**

Strengths:

- GPU-accelerated differentiable pipeline to solve Poisson equation could be useful to research community at large

- New shape parameterisation has similar properties to neural implicit fields (can handle arbitrary topology, produces watertight surfaces) while being more lightweight and efficient

- Extensive experimental evaluation demonstrate the advantages of the proposed representation when 3D supervision is available

Weaknesses:

- Limited resolution: the proposed pipeline is, by design, limited by the cubic memory requirement at training time. This is not the case for neural implicit fields, and in some sense the proposed representation is a step-back from them, which allowed to represent arbitrary topology while breaking free from memory intensive 3D grids.

- 2D supervision: the proposed approach is not evaluated in a context in which 3D supervision is not available, although, in theory, any off the shelf differentiable renderer would be compatible with the pipeline. Without this evaluation, it is hard to say if the proposed approach has any fundamental limitation in a differentiable rendering context (e.g. due to its underlying sparsity?).

Question for the authors:

along lines 45~48 authors claim: "It is easy to initialize SAP with a given geometry such as template shapes or noisy observations. In contrast, neural implicit representations are harder to initialize, except for few simple primitives like spheres". The claim is re-stated in table 1. I don't understand what do authors exactly mean with this:
In the learning setting, where oriented clouds of points are used as underlying 3D shape representation, we do not really care about initialisation. The same though holds for neural implicit fields, as the weights of the underlying MLP are similarly initialised using standard techniques.
In the optimization setting, although I understand that the SAP framework would allow for initialising the shape using noisy points and normals, the reported experiments still decide to start from a sphere (see supplemental video), probably because this results in a better local minima attained by the optimisation process. Note that a sphere is indeed a trivial initialisation for  an implicit field, as correctly stated by the authors along lines 45~48.
In light of all this, I think this claim should be either better argumented/explained or toned down.

**Time Spent Reviewing:**

4

---

> ### Author Response · Authors · 2021-08-09
> **Response to Reviewer eo1g**
>
> We thank the reviewer for considering our method useful to the research community at large. We address the remaining comments below.
>
> **Limited by the cubic memory requirement, a step back from neural implicit fields**
>
> Thanks for the comment. Indeed, we also point out cubic memory requirements as one limitation in the conclusion part. Nevertheless, we believe that future improvements using space-adaptive data structures (e.g., octrees) can significantly alleviate this issue during training. During inference, processing scenes in a sliding-window manner (akin to Convolutional Occupancy Networks) will enable the application of our method to larger scenes. We identify this as promising future work. Moreover, although neural-implicit methods do not require dense grid evaluation during training, they do require dense evaluation during inference, which leads to slow inference and necessitates hierarchical approaches like Multiresolution IsoSurface Extraction (MISE). In contrast, our method can reconstruct at significantly faster speeds, c.f., Table 4 in the main paper.
>
> **No evaluation on only 2D supervision so it is hard to say if the approach has any fundamental limitation in a differentiable rendering context**
>
> Our primary goal was to validate the effectiveness of our shape-as-points representation. Therefore, we follow the evaluation setting of existing methods including 3D-R2N2, AtlasNet and ConvONet which use 3D supervision. Nevertheless, we thank the reviewer for the great suggestion which inspired us to do a preliminary proof-of-concept experiment for the task of 3D reconstruction from multi-view images. Indeed, our differentiable point-to-mesh layer under the optimization-based setting can be naturally integrated with a differentiable renderer.
>
> We proceed as follows: After obtaining the mesh in each iteration, for each 2D image, we rasterize the 3D mesh using `rasterize_meshes` in PyTorch3D to find the corresponding surface points for each pixel, and use an MLP to predict their RGB colors. An L2 loss is applied to the predicted and input RGB images. In addition, we also apply a silhouette loss using a differentiable silhouette renderer in PyTorch3D. We have uploaded some preliminary results to https://imgur.com/wu210KX. We tested our implementation on 1 synthetic object (cow) and 2 real-world objects in DTU dataset (skull and bird). As can be observed, our method is indeed able to reconstruct detailed geometry using only 2D supervision, and we consider this as an exciting extension for future work.
>
> We further remark that recent methods based on neural implicit representations (e.g. [DVR, CVPR’20]) can attain high-quality 3D reconstruction, but their optimization process is slow due to dense evaluation in the ray marching step. In contrast, in each iteration our method outputs an explicit mesh. Therefore, no expensive ray marching is required to find the surface, leading to faster inference.
>
> **Not understand the “easy to initialize” claim**
>
> Thanks for the question. In the learning setting, what we want to emphasize is geometric initialization instead of network initialization. For neural-implicit methods, weight initializations can only be analytically derived for simple shapes like spheres. In contrast, since our networks only need to predict the offset and normal fields for the input point cloud, this input point cloud is directly treated as the geometric initialization. In the table below, we show the Chamfer distance evaluated in the ShapeNet test set with 3K input points (noise level=0.005) at different training iterations. As can be seen, our method enables much faster training convergence than ConvONet (neural-implicit method), which illustrates the effectiveness of the geometric initialization directly from input point clouds.
>
> |Iterations|10k|50k|100k|200k|Best model|
> |-|-|-|-|-|-|
> |ConvONet|0.082|0.058|0.055|0.050|0.044|
> |Ours|0.041|0.036|0.035|0.034|0.034|
>
> In the optimization setting, we indeed chose to start from a sphere, since we intended to demonstrate that if our method is able to produce decent 3D reconstruction even starting from a sphere, we can also faithfully reconstruct from a coarse or noisy shape since it is a simpler task, and it will converge faster. We kindly refer the reviewer to https://imgur.com/BejFVcB for a comparison of different initializations. As can be observed, when starting from points+normals sampled from a coarse shape, our method converges faster. In terms of accuracy, both results are equivalent, i.e., there is no “better local minima attained by the optimisation process”. We will add these results and a discussion to the supplementary.

---

> > ### Comment · Reviewer_eo1g · 2021-09-10
> > **post rebuttal**
> >
> > Thanks for the clarification. The DR results look very promising, and perhaps you should share them with the community in the Supplementary material! I am going to upgrade my score in light of all this.

---

### Official Review · Reviewer_QqU1 · 2021-07-14

**Rating:** 9
**Confidence:** 4

**Summary:**

This work proposes a novel shape representation using point clouds. This is enabled by unique differentiable poisson solver layer that enables to convert the oriented point clouds to full volume representing the shape. I think the main contribution of this work is the spectral technique that solves poisson surface reconstruction problem, which can be efficiently integrated into neural workflows. While this work mostly focuses on surface reconstruction, I believe this representation can become broadly applicable to various problem domains.

**Limitations And Societal Impact:**

Limitations are adequately addressed.

**Main Review:**

This paper has both high-level novelty ("shape-as-points" representation) as well as low-level technical contribution (spectral differentiable poisson solver). I think the proposed representation is valuable and will inspire future work. The evaluation is thorough and the method offers several quantitative advantages over state of the art methods (in terms of runtime and quality). I would be excited to see this work at NeurIPS.

**Time Spent Reviewing:**

1

---

> ### Author Response · Authors · 2021-08-09
> **Response to Reviewer QqU1**
>
> We thank the reviewer for acknowledging the novelty of our paper, and pointing out that the proposed representation is broadly applicable to various problem domains. We are equally excited about the prospects of this methodology, as well as its applications to various other important problems in the future.

---

### Official Review · Reviewer_nyEb · 2021-07-15

**Rating:** 8
**Confidence:** 4

**Summary:**

This paper proposes a differentiable adaptation of the Poisson Surface Reconstruction algorithm for oriented point sets. It discretizes the Poisson equation over a 3D grid and use spectral methods to find the solution for indicator function values over this 3D grid. The final grid is fed to marching cubes algorithm to obtain the final mesh. The differentiability of the approach allows to use the proposed method in various optimization-based applications from which the authors consider optimization-based and learning-based 3D reconstruction tasks. Proposed models are evaluated on datasets with different level of details and noise present in the inputs and compared to both classical and more recent learning-based methods.

**Limitations And Societal Impact:**

The authors do not provide any discussions about these impacts besides stating that their approach shares the risks with all the other learning-based 3D reconstruction approaches.

**Main Review:**

Pros:
* The paper is well-written.
* The proposed approach is novel and promising, both qualitative and quantitative results shown are strong.
* Evaluations are extensive and include both different datasets and different noise injection configurations. The authors additionally provide evaluations of the efficiency of the proposed method as well as some additional ablation studies.

Cons:
* The choice of some baselines for learning-based approaches is questionable. None of PSGN, 3D-R2N2 and AtlasNet models were designed for surface reconstruction from input unoriented points, so they were not optimized for this task. There is a least one very relevant work left unmentioned and uncompared to in this submission [1].
* It is clear that it is hard to provide a full mathematical explanation in the main text without using too much space but even in the supplementary materials some detail are missing. My concern is mainly equation 7 from the supplementary. Equations 5 and 6 partly justify it but do not account for the appearance of $\tilde{g}$. It is said that it is introduced to mitigate Gibbs phenomenon, but it is not obvious why the addition of $\tilde{g}$ to the solution of Poisson equation in frequency domain do not corrupt this solution (there will not be an identity if one put $\tilde{\chi}$ in FFT of Poisson equation).

Overall, I believe this is a strong submission. The cons should be addressed but, in fact, are minor in my opinion and should not justify rejection.

Minor comments:
1) L146-148 "$\tilde{g}_{\sigma, r}(u)$..." - rephrase the sentence, something is missing.
2) L183 "a a"

[1] Erler, P., Guerrero, P., Ohrhallinger, S., Mitra, N. J., Wimmer, M. Points2surf: learning implicit surfaces from point clouds. In ECCV’20.

***
After reading the rebuttal and other reviews I decided to improve my rating since my concerns were answered. This is an overall strong submission with great novelty and promising experimental results despite the limitations of cubic complexity for indicator grids.

**Time Spent Reviewing:**

5

---

> ### Author Response · Authors · 2021-08-09
> **Response to Reviewer nyEb**
>
> We thank the reviewer for the constructive feedback. We appreciate that the reviewer finds our paper promising, novel, and well-written. We address additional comments below.
>
> **None of PSGN, 3D-R2N2 and AtlasNet models were designed for surface reconstruction from input unoriented points, so they were not optimized for this task**
>
> We propose SAP as a novel shape representation. Therefore, we find it important to compare SAP against different established shape representations with PSGN representing point clouds, 3D-R2N2 voxels and AtlasNet meshes. As all of these methods are encoder-decoder approaches, this comparison provides a fair analysis of the quality of each output representation. We further remark that other related works (e.g., Occupancy Networks) use a similar evaluation protocol.
>
> **Compare to Points2Surf, ECCV’20**
>
> We thank the reviewer for the suggestion. For this rebuttal, we evaluated Points2Surf on the ShapeNet test set with 3K input points (noise level=0.005) and obtained the following results which we will include in the paper:
>
> ||Chamfer-L1|F-Score|Normal C.|
> |-|-|-|-|
> |Points2Surf|0.068|0.808|0.857|
> |Ours|**0.034**|**0.975**|**0.944**|
>
> Note that Points2Surf requires roughly 75 seconds for inference of a single shape at a resolution of $128^3$, while our method requires only 0.07 seconds.
>
> Remark: As training of Points2Surf requires over one week on 4 RTX 2080Ti and since the authors mention that “Points2Surf is patch-based and therefore independent from classes”, we use the model provided by the authors for this experiment. For the final version of the paper we will also add the results of a model retrained on our data.
>
> **Concern is mainly equation 7 from the supplementary. Equations 5 and 6 partly justify it but do not account for the appearance of $\tilde{g}$. It is said that it is introduced to mitigate Gibbs phenomenon, but it is not obvious why the addition of $\tilde{g}$ to the solution of Poisson equation in frequency domain do not corrupt this solution (there will not be an identity if one put $\tilde{\chi}$ in FFT of Poisson equation)**
>
> 1. *Why we need a Gaussian*: The Gaussian serves as a regularizer to the smoothness of the solved implicit function. Not using a Gaussian is equivalent to using a Gaussian kernel with sigma = 0. Please refer to Fig. 3 in the supplementary which motivates the use of our sigma parameter. We will rephrase Eqn. 5, 6, 7 in the supplementary to further discuss and motivate the inclusion of the Gaussian term.
>
> 2. *Why using a Gaussian in the spectral domain*: First, the FFT of a Gaussian remains a Gaussian. Second, convolution of a Gaussian in the physical domain is equivalent to a dot product with a Gaussian in the spectral domain and a dot product in the spectral domain is more efficient than convolution in the physical domain: $O(NlogN)$ vs $O(N^2)$, where $n$ is the resolution of a regular grid and $N = n^3$. We will clarify this in the paper.
>
> **Minor comments**
>
> Thanks for pointing out the typos, we will update the final version accordingly.

---

### Decision · Program_Chairs · 2021-09-27

**Decision:**

Accept (Oral)

**Comment:**

Congratulations, the paper is accepted to NeurIPS 2021!
Please incorporate additional experiments, edits and corrections as discussed in rebuttal.